# DiffusionReward: Enhancing Blind Face Restoration through Reward Feedback Learning

## Abstract

Reward Feedback Learning (ReFL) has recently shown great potential in aligning model outputs with human preferences across various generative tasks. In this work, we introduce a ReFL framework, named *DiffusionReward*, to the Blind Face Restoration task for the first time. DiffusionReward effectively overcomes the limitations of diffusion-based methods, which often fail to generate realistic facial details and exhibit poor identity consistency. The core of our framework is the Face Reward Model (FRM), which is trained using carefully annotated data. It provides feedback signals that play a pivotal role in steering the optimization process of the restoration network. In particular, our ReFL framework incorporates a gradient flow into the denoising process of *off-the-shelf* face restoration methods to guide the update of model parameters. The guiding gradient is collaboratively determined by three aspects: (i) the FRM to ensure the perceptual quality of the restored faces; (ii) a regularization term that functions as a safeguard to preserve generative diversity; and (iii) a structural consistency constraint to maintain facial fidelity. Furthermore, the FRM undergoes dynamic optimization throughout the process. It not only ensures that the restoration network stays precisely aligned with the real face manifold, but also effectively prevents reward hacking. Experiments on synthetic and wild datasets demonstrate that our method outperforms state-of-the-art methods, significantly improving identity consistency and facial details. The source codes and models are available at: https://anonymous.4open.science/r/DiffusionReward-D02F

## 1 Introduction

Facial images captured in-the-wild often suffer from complex and diverse degradations, such as blur, compression artifacts, noise, and low resolution. Blind Face Restoration (BFR) (Li et al., 2018; 2020; Wang et al., 2021) aims to restore high-quality (HQ) counterparts from these degraded inputs. Given the substantial information loss in low-quality (LQ) inputs and the typically unknown degradation processes, BFR is inherently a highly ill-posed problem. As a result, for any given single LQ face, there theoretically exists a solution space encompassing an infinite number of potential high-quality solutions. Consequently, accurately reconstructing HQ facial images from this expansive solution space remains an unsolved challenge, especially in terms of photorealism, naturalness, and identity preservation.

Diffusion models (Ho et al., 2020) have become a powerful paradigm for BFR (Wu et al., 2024; Lin et al., 2024; Chen et al., 2024; Yue & Loy, 2024; Wang et al., 2023b), owing to their exceptional generative capabilities. Using rich visual priors acquired during training, these models use LQ images as conditional inputs to progressively reconstruct high-fidelity faces through iterative denoising. Notable methods, such as DiffBIR (Lin et al., 2024) and OSEDiff (Wu et al., 2024), leverage the pre-trained Stable Diffusion (Rombach et al., 2022) models, effectively adapting them through fine-tuning to achieve remarkable quality in face restoration. However, these pre-trained diffusion models typically undergo training using images from general domains, which lack an adequate amount of face-specific prior knowledge. This deficiency frequently gives rise to restored facial images that are short of detailed features.

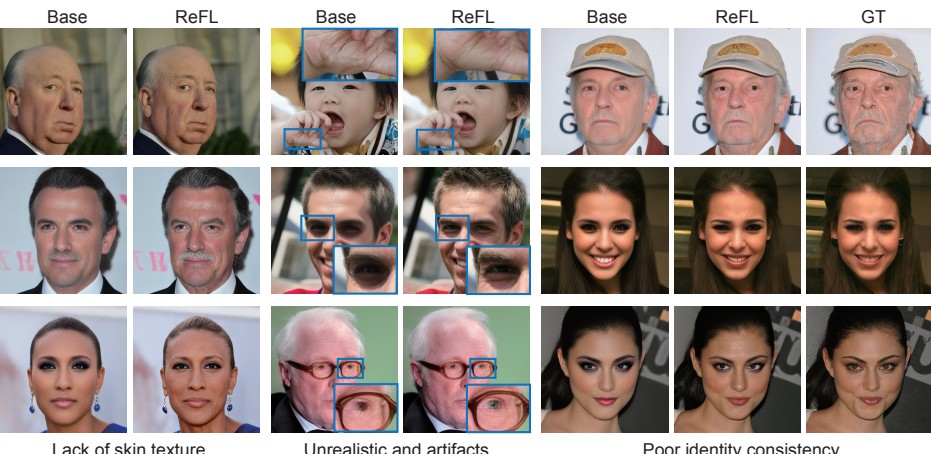

Figure 1: An example of issues with diffusion-based face restoration methods. After enhancement with ReFL, the issues in the base model are significantly mitigated.

As shown in Figure 1 (Left), although coarse facial features, accessories, and background areas can be restored to a reasonable extent, the restoration of fine-grained facial textures, such as skin textures, is usually insufficient, leading to overly smooth or unrealistic textures (Zhang et al., 2025). The lack of face-specific priors not only undermines the restoration quality of fine details but also significantly exacerbates mapping ambiguities (Kamali et al., 2025), as shown in Figure 1 (Middle). Furthermore, Stable Diffusion models are primarily trained for text-to-image generation tasks, rather than for image restoration tasks which requires strict fidelity. Consequently, their inherent generative mechanisms and the nature of the training data are more adept at creative synthesis rather than meeting the exacting standards of fidelity demanded by restoration tasks, potentially leading to deviations from the original identity features during the restoration process, as shown in Figure 1 (Right).

Reward Feedback Learning (ReFL) (Xu et al., 2023; Clark et al., 2023; Liang et al., 2024) is an optimization paradigm that has been validated in domains such as text-to-image generation. It makes use of a reward model that has been trained based on human preferences. This reward model serves to guide and fine-tune latent diffusion models, boosting the quality, realism, and user alignment of the outputs generated by these models. In this work, we employ ReFL for the BFR task to address the previously mentioned limitations of diffusion-based face restoration methods.

For *off-the-shelf* diffusion-based face restoration methods (Lin et al., 2024; Wu et al., 2024), the ReFL framework innovatively reinterprets their latent diffusion denoising process as a parameterized iterative generator. Through the parameterization of this process, ReFL empowers the application of supplementary optimization constraints. This enables fine-grained adjustments to the parameters of pre-trained face restoration models. Consequently, fine-tuned models are capable of generating images that feature enhanced facial texture details, a higher level of overall visual realism, and, more importantly, the preservation of identity consistency. A core component of the ReFL framework is a reward model that is able to accurately assess image quality.

To this end, we have meticulously annotated the data and constructed a Face Reward Model (FRM). This model serves as a crucial component for evaluating the quality of restored faces. It provides feedback signals that play a pivotal role in steering the optimization process of the face restoration model. One common challenge in the training process based on ReFL is that the restoration model might fall prey to reward hacking. It occurs when the restoration model discovers and capitalizes on "loopholes" within the reward model instead of enhancing the actual perceptual quality of the images. To address this issue, we further propose a strategy for dynamically updating the FRM during the training process. In this manner, the reward model can continuously adapt to the evolution of the restoration model, thereby more precisely guiding its exploration and optimization within the manifold space of real facial images, effectively averting the phenomenon of overfitting to a specific reward function.

In addition, we also introduce two constraints to further enhance the restoration performance. Firstly, a Structural Consistency Constraint is incorporated to ensure that the restored image's facial structure

closely aligns with the original identity, thereby effectively preserving identity consistency. By doing so, it effectively safeguards the identity consistency, preventing any significant discrepancies in the facial features. Secondly, a Weight Regularization term is employed to restrict the extent to which the current model parameters deviate from their initial values. Through this mechanism, it maintains the inherent generative capabilities of the base model, ensuring that the output diversity is not compromised.

In summary, here are our main contributions:

- We make a pioneering exploration into the BFR domain by introducing ReFL, crafting a bespoke ReFL optimization mechanism designed specifically for diffusion-based face restoration models.
- We tailor a data curation pipeline for the creation of an FRM that is capable of accurately evaluating the perceptual quality of restored facial images. Moreover, we introduce a dynamic updating strategy to avert the reward hacking problem.
- We introduce two constraints to further enhance the restoration performance, including a structural consistency constraint and a weight regularizer.
- Our proposed framework, named *DiffusionReward*, enhances the face restoration quality of the base model and achieves state-of-the-art (SOTA) performance compared to other advanced methods.

## 2 RELATED WORK

**Blind Face Restoration.** Early Blind Face Restoration (BFR) methods mainly relied on geometric priors to provide structural guidance. These include 2D priors such as facial landmarks (Chen et al., 2018; Kim et al., 2019), parsing maps (Chen et al., 2021; Shen et al., 2018), and component heatmaps (Yu et al., 2018), as well as 3D facial priors (Hu et al., 2020) which explicitly utilize 3D morphable models to grasp sharp facial structures. However, these geometric priors exhibit limitations in recovering fine-grained details, like skin textures, and struggle with severely degraded inputs.

Generative facial priors have emerged as a significant pathway for high-quality face restoration (Ledig et al., 2017; Wang et al., 2018). Pre-trained StyleGAN (Karras et al., 2019; 2020), encapsulating rich facial textures and details, facilitate photorealistic face restoration. For instance, GFP-GAN (Wang et al., 2021) and GLEAN (Chan et al., 2021) integrate StyleGAN priors into an encoder-decoder architecture, leveraging structural features from degraded faces to guide restoration, thereby remarkably enhancing detail recovery. However, degraded inputs may be mapped to suboptimal points within the latent space, leading to insufficient fidelity or undesirable artifacts. Codebook-based methods (Gu et al., 2022; Zhou et al., 2022; Zhao et al., 2022) employ vector-quantized codebooks to mitigate latent space uncertainty by learning discrete priors. Among them, Zhao et al. (Zhao et al., 2022) incorporated these discrete priors into skip connections to enhance reconstruction fidelity, while simultaneously injecting adaptive stochastic noise to improve generation quality.

Denoising Diffusion Probabilistic Models (DDPMs) (Sohl-Dickstein et al., 2015; Ho et al., 2020) have recently become an emergent paradigm in BFR, due to their powerful generative capabilities and training stability. DR2 (Wang et al., 2023b) initially generates a coarse output by noising and subsequently denoising the degraded face, which is then refined by other face restoration models for detail enhancement. DiffBIR (Lin et al., 2024) decouples BFR into two distinct stages: degradation removal and generative refinement. In the degradation removal stage, advanced restoration modules such as SwinIR (Liang et al., 2021) are employed. Subsequently, in the generative refinement, an IRControlNet (Lin et al., 2024) is utilized to guide a latent diffusion model for detail generation. DiffFace (Yue & Loy, 2024) constructs a posterior distribution from low-quality (LQ) to high-quality (HQ) images, leveraging the error-shrinkage property of pre-trained diffusion models to robustly handle unknown degradation.

Despite the strengths of diffusion-based methods, their multi-step sampling process often leads to slower inference. To enhance inference efficiency, several diffusion-based image restoration methods employing distillation for one-step inference have emerged. Notably, OSEDiff (Wu et al., 2024) fine-tunes Stable Diffusion (Rombach et al., 2022) using variational score distillation, achieving high-quality restoration with one-step inference. In this work, to validate the generalizability of our method across diffusion-based methods, we choose OSEDiff and DiffBIR as base models, embodying single-step and multi-step diffusion paradigms, respectively.

**Reward Feedback Learning.** In the text-to-image (T2I) generation with ReFL field, there are two primary stages. Initially, a reward model is trained using human preference data, such as

pairwise comparisons or ratings, to capture and quantify human preferences like perceptual image quality, text-image alignment, and other aesthetic criteria. Subsequently, the trained reward model guides the optimization of the T2I model by leveraging gradients derived from its scores. Previous work (Xu et al., 2023; Kirstain et al., 2023; Liang et al., 2024; Zhang et al., 2024) have constructed preference datasets and corresponding reward models for T2I tasks. Moreover, some studies have explored the potential of leveraging feedback derived from reward models to effectively optimize T2I models. ImageReward (Xu et al., 2023) evaluates images predicted at specific denoising steps and backpropagates gradients from these scores to directly fine-tune the diffusion model parameters. In contrast, DRaFT (Clark et al., 2023) and AlignProp (Prabhudesai et al.) evaluate only the final image for optimization. R0 (Luo et al., 2025) achieves state-of-the-art T2I by directly maximizing rewards without complex diffusion losses. While existing ReFL paradigms succeed in open-ended text-to-image synthesis, their direct application to image restoration is constrained by the precise face assessment and strict identity maintenance. We overcome these limitations by incorporating two key refinements to the ReFL framework: (i) a specialized Face Reward Model (FRM) for accurate facial quality assessment, and (ii) an structural consistency constraint to enforce identity preservation. Furthermore, we implement an innovative dynamic updating mechanism to effectively mitigate reward hacking, thereby yielding a substantial elevation in overall restoration quality.

## 3 DIFFUSIONREWARD

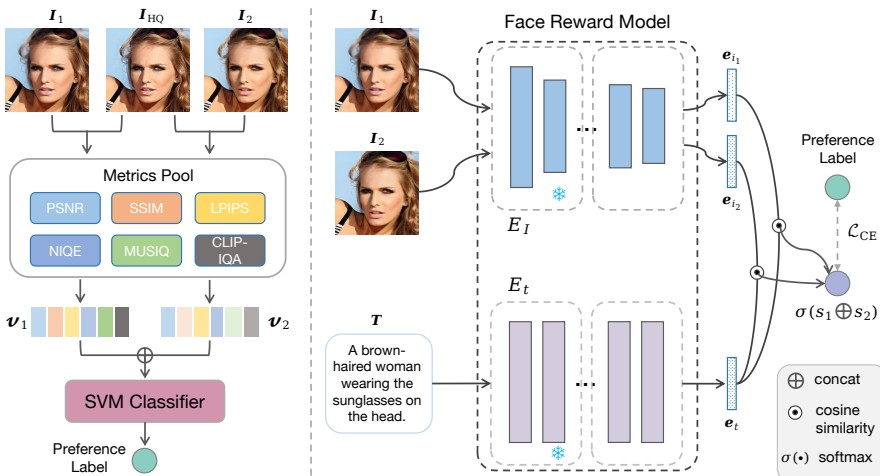

Figure 2: Training framework of the Face Reward Model. We first train a SVM (Cortes & Vapnik, 1995) classifier for automated annotation. The classifier is trained with the metric vectors ($\boldsymbol{v}_1$, $\boldsymbol{v}_2$) and annotated supervision signals (Left). The face reward model is based on the CLIP (Radford et al., 2021) architecture (Right), where the last 20 layers of the image encoder $E_I$ and the last 11 layers of the text encoder $E_t$ are trainable, while the remaining parameters are frozen. $s_1$ and $s_2$ are derived from the similarity between the image embedding and the text embedding (e.g., $< \boldsymbol{e}_{i_1}, \boldsymbol{e}_t >$).

### 3.1 FACE REWARD MODEL

General-purpose reward models, which are commonly trained on human ratings of natural or artistic images, incorporate only limited face image ratings, leading to significant biases in providing reliable and accurate evaluations for face-related restoration. To tackle this issue, we design a pipeline for constructing a face reward model, which consists of two essential stages: annotation of a preference dataset and training of the face reward model.

**Annotation of the Preference Dataset.** To construct the face preference dataset, we select 19,590 diverse face images from the face dataset (Wu et al., 2023b), encompassing various poses and expressions. Then, we use LLaVA (Liu et al., 2023) to generate corresponding textual descriptions for each image, forming 19,590 image-text pairs. Subsequently, we apply blind degradation kernels (See details in Section (4.1)) to the high-quality images $\mathbf{I}_{\text{HQ}}$, producing their low-quality (LQ) counterparts $\mathbf{I}_{\text{LQ}}$. We employed three blind face restoration methods (Zhou et al., 2022; Lin et al., 2024; Chan et al., 2021) to restore these LQ images, yielding a total of 58,770 ($3 \times 19,590$) restored face images.

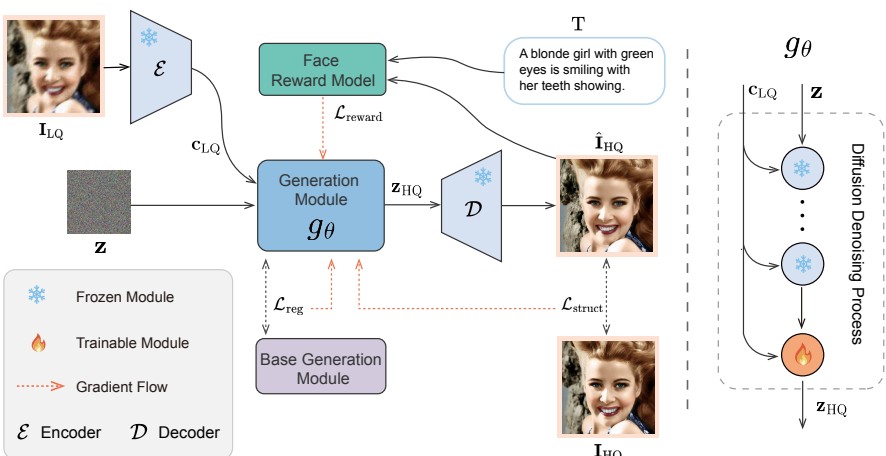

Figure 3: Our ReFL training framework. (Left) We introduce multiple constraints to optimize the generation module $g_\theta$, including $\mathcal{L}_{\text{reward}}$, $\mathcal{L}_{\text{reg}}$ and $\mathcal{L}_{\text{struct}}$ (See details in Section 3.3). (Right) For training efficiency, these constraints are applied solely on the last denoising step.

Finally, these restored images, combined with the original 19,590 ground-truth images, constitute our preference dataset of 78,360 ($4 \times 19,590$) facial images, providing a comprehensive data base for subsequent preference annotation.

Given an original facial image $\mathbf{I}_{\text{HQ}}$ and its counterparts of three restored versions $\{\mathbf{I}_1, \mathbf{I}_2, \mathbf{I}_3\}$, we conduct pairwise comparisons among these images that yield six preference pairs. In the annotation phase, any preference pair involving the $\mathbf{I}_{\text{HQ}}$ was assigned a fixed label indicating a preference for the ground-truth image, thereby treating the $\mathbf{I}_{\text{HQ}}$ as an ideal and optimal result. The remaining preference pairs, which involved comparisons between different restoration results, are labeled using a hybrid strategy by combining human manual annotation and automated annotation.

Fully relying on human annotation would be prohibitively costly. To address this problem, we developed an efficient hybrid annotation strategy. Human annotators label a subset of image pairs (Refer to Appendix A.1 for annotation details), while the remaining pairs are automatically labeled by a preference predictor, as illustrated in Figure 2 (Left). For each pair of images, we compute six evaluation metrics: SSIM (Wang et al., 2004), PSNR, LPIPS (Zhang et al., 2018), MUSIQ (Ke et al., 2021), NIQE (Mittal et al., 2012), and CLIP-IQA (Wang et al., 2023a). These metrics are then vectorized (*i.e.*, $\boldsymbol{v}_1$ and $\boldsymbol{v}_2$ in Figure 2) and fed into a annotation predictor. The SVM (Cortes & Vapnik, 1995) classifier is trained using human-annotated preference labels. With the classifier, the remaining preference pairs are automatically annotated, significantly reducing annotation costs. The detailed configuration and hyperparameters of the SVM classifier can be found in Appendix A.1.

**Reward Model Training.** Training a reward model from scratch is inefficient. Instead, we fine-tuned the pre-trained HPSv2 model (Wu et al., 2023a), which is based on the CLIP architecture (Radford et al., 2021) and pre-trained on large-scale image datasets, providing robust image quality assessment priors suitable for adaptation to face preference data. We fine-tune HPSv2 with the 117,540 preference image-text pairs to optimize its ability to predict the relative quality of face images, and the training process is illustrated in Figure 2 (Right). For training efficiency, we set the last 20 layers of the image encoder and the last 11 layers of the text encoder trainable, while freeze the remaining parameters.

Given the restored images $\mathbf{I}_1$ and $\mathbf{I}_2$, we can collect their corresponding embeddings $\boldsymbol{e}_{i_1}$ and $\boldsymbol{e}_{i_2}$ through the same image encoder $E_I$. Then, we use the text encoder $E_t$ to represent the input text $\mathbf{T}$ as $\boldsymbol{e}_t$. Next, we calculate $s_1$ and $s_2$ that refer to the cosine similarities between $\boldsymbol{e}_{i_1}$-$\boldsymbol{e}_t$ and $\boldsymbol{e}_{i_2}$-$\boldsymbol{e}_t$, respectively. subsequently, $s_1$ and $s_2$ are concatenated and followed by a softmax operation as the probabilities of preference. Finally, we minimize the entropy loss $\mathcal{L}_{\text{CE}}$ between the preference label, derived from the SVM classifier combined with human annotations, and the probabilities $\sigma([s_1; s_2])$. During the inference stage, the reward model only requires an input image and its corresponding text description to calculate the preference score, thereby completing the evaluation of image quality.

## 3.2 Modeling the Denoising Process

We develop on Stable Diffusion (Rombach et al., 2022) models for the BFR task. Using the pretrain autoencoder (Kingma et al., 2013; Rombach et al., 2022), we convert the $\mathbf{I}_{\mathrm{HQ}}$ into a latent $\mathbf{z}_{\mathrm{HQ}}$ with image encoder $\mathcal{E}$ (*i.e.*, $\mathbf{z}_{\mathrm{HQ}} = \mathcal{E}(\mathbf{I}_{\mathrm{HQ}})$) and reconstruct it with decoder $\mathcal{D}$ (*i.e.*, $\hat{\mathbf{I}}_{\mathrm{HQ}} = \mathcal{D}(\mathbf{z}_{\mathrm{HQ}})$). Both diffusion and denoising process, Gaussian noise with variance $\beta_t \in (0, 1)$ at time $t$ is added to the encoded latent $\mathbf{z}_{\mathrm{HQ}}$ to produce the noisy latent: $\mathbf{z}_t = \sqrt{\bar{\alpha}_t}\mathbf{z}_{\mathrm{HQ}} + \sqrt{1 - \bar{\alpha}_t}\boldsymbol{\epsilon}$, where $\boldsymbol{\epsilon} \sim \mathcal{N}(0, \mathbf{I})$, $\alpha_t = 1 - \beta_t$ and $\bar{\alpha}_t = \prod_{s=1}^{t} \alpha_s$. When $t$ is large enough, the latent $\mathbf{z}_t$ is close to a standard Gaussian distribution. A network $g_\theta$ is learned by predicting the noise $\epsilon$ conditioned on $\mathbf{c}_{\mathrm{LQ}} = \mathcal{E}(\mathbf{I}_{\mathrm{LQ}})$ at a random time-step $t$.

As shown in Figure 3, the denoising process of the face restoration facilitates the subsequent introduction of gradient information to optimize the parameters of the restoration model. Thus, this conditional denoising process can be interpreted as a parameterized generation module $g_\theta(\mathbf{z}_t, \mathbf{c}_{\mathrm{LQ}}, t)$ in the latent space. The training objective for the base restoration model is a noise prediction loss:

$$\mathcal{L}_{\mathrm{ldm}} = \mathbb{E}_{\mathbf{z}, \mathbf{c}_{\mathrm{LQ}}, t, \boldsymbol{\epsilon}}[\|\boldsymbol{\epsilon} - g_\theta(\sqrt{\bar{\alpha}}\mathbf{z} + \sqrt{1 - \bar{\alpha}_t}\boldsymbol{\epsilon}, \mathbf{c}_{\mathrm{LQ}}, t)\|_2^2]. \tag{1}$$

This objective is solely utilized for pretraining the off-the-shelf diffusion-based BFR models (Lin et al., 2024; Wu et al., 2024); our reward-based fine-tuning objective, applied to the final restored image, is detailed in Section 3.3.

Within this framework, different BFR methods vary in the specific implementation of the denoising network $g_\theta$ and its utilization of the conditions $\mathbf{c}_{\mathrm{LQ}}$. For multi-step inference models like DiffBIR (Lin et al., 2024), $g_\theta$ refers to a UNet (Ronneberger et al., 2015) with ControlNet (Zhang et al., 2023). Its initial input is the primarily noise $\mathbf{z}$, and the condition $\mathbf{c}_{\mathrm{LQ}}$ is integrated to each denoising step. For single-step inference models like OSEDiff (Wu et al., 2024), $\epsilon_\theta$ refers to a UNet with a LoRA (Hu et al., 2022) module. The condition $\mathbf{c}_{LQ}$ is directly injected to the initial noise $\mathbf{z}$ by a concatenation operation. Thus, it eliminates the need for iterative injection.

## 3.3 ReFL: Training Objectives and Strategies

We introduce three additional objective functions, including reward loss, structural consistency loss, and weight regularization loss, to refine the generation module $g_\theta$ for better perceptual quality and identity consistency of restored faces, as shown in Figure 3.

**Reward Loss.** To enhance the alignment with human preference on the restored faces, we leverage the pre-trained face reward model $\mathcal{R}$ (See Section 3.1) to provide assessment feedbacks. The face reward model takes the restored image $\hat{\mathbf{I}}_{\mathrm{HQ}}$ and the text description $\mathbf{T}$ of corresponding original image $\mathbf{I}_{\mathrm{HQ}}$ as input, where $\hat{\mathbf{I}}_{\mathrm{HQ}}$ is obtained by decoding the latent of the last denoising step: $\hat{\mathbf{I}}_{\mathrm{HQ}} = \mathcal{D}(\mathbf{z}_{\mathrm{HQ}})$. Thus, the reward loss $\mathcal{L}_{\mathrm{reward}}$ is defined as:

$$\mathcal{L}_{\mathrm{reward}} = -\mathcal{R}(\hat{\mathbf{I}}_{\mathrm{HQ}}, \mathbf{T}). \tag{2}$$

By minimizing $\mathcal{L}_{\mathrm{reward}}$, we encourage $g_\theta$ to generate restored faces with higher alignment scores with human preference.

**Structural Consistency Loss.** To maintain high fidelity to the structural features of real faces and improve identity consistency, we introduce both structural and perceptual level constraints, which comprises two sub-components:

- *LPIPS Loss:* LPIPS (Zhang et al., 2018) is a highly prevalent metric for evaluating the perceptual similarity between two input images. Unlike traditional pixel-wise metrics (*e.g.*, MSE, PSNR), LPIPS leverages deep neural networks to extract hierarchical semantic features from images, aligning more closely with human visual perception. We employ the LPIPS to measure the perceptual similarity between $\hat{\mathbf{I}}_{\mathrm{HQ}}$ and the original image $\mathbf{I}_{\mathrm{HQ}}$:

$$\mathcal{L}_{\mathrm{LPIPS}} = \mathrm{LPIPS}(\hat{\mathbf{I}}_{\mathrm{HQ}}, \mathbf{I}_{\mathrm{HQ}}). \tag{3}$$

- *DWT Low-Frequency Loss:* Given the pixel-wise losses (*e.g.*, $\ell_1$, MSE) are limited in boosting the vivid and intricate details, we apply Discrete Wavelet Transform (DWT) to ensure the low-frequency components of the restored image consistent to the original image. Moreover, we constrain only the low-frequency components of the image (*i.e.*, better structural consistency), allowing the restoration

model to explore Freely in the high-frequency components (*i.e.*, better details). Let $\text{DWT}_{\text{LF}}(\cdot)$ denote the function that extracts low-frequency components; the $\mathcal{L}_{\text{DWT}}$ is defined as:

$$\mathcal{L}_{\text{DWT}} = \|\text{DWT}_{\text{LF}}(\mathcal{D}(z_{\text{HQ}})) - \text{DWT}_{\text{LF}}(\mathbf{I}_{\text{HQ}})\|_1. \tag{4}$$

**Weight Regularization Loss.** To prevent the parameters $\boldsymbol{\theta}$ in $g_\theta$ from deviating excessively from its initial state $\boldsymbol{\theta}_{\text{base}}$ (*e.g.*, pre-trained weights of the diffusion models), we incorporate a regularization term of Kullback–Leibler divergence:

$$\mathcal{L}_{\text{reg}} = \mathcal{D}_{\text{KL}}(\boldsymbol{\theta}\|\boldsymbol{\theta}_{\text{base}}). \tag{5}$$

The final objective is a weighted combination:

$$\mathcal{L}_{\text{total}} = \lambda_{\text{reward}}\mathcal{L}_{\text{reward}} + \lambda_{\text{LPIPS}}\mathcal{L}_{\text{LPIPS}} + \lambda_{\text{DWT}}\mathcal{L}_{\text{DWT}} + \lambda_{\text{reg}}\mathcal{L}_{\text{reg}}. \tag{6}$$

where $\lambda_{\text{reward}}$, $\lambda_{\text{LPIPS}}$, $\lambda_{\text{DWT}}$ and $\lambda_{\text{reg}}$ are balancing hyperparameters. The parameters $\boldsymbol{\theta}$ of $g_\theta$ are updated by minimizing $\mathcal{L}_{\text{total}}$ during ReFL fine-tuning. At each iteration, we obtain $\hat{\mathbf{I}}_{\text{HQ}}$ via the full reverse denoising trajectory ($z_T \to z_0$). However, to ensure efficiency, we employ truncated backpropagation (Clark et al., 2023) rather than computing gradients through the entire chain. Gradients of $\mathcal{L}_{\text{total}}$ are propagated only through the last $N$ denoising steps of $g_\theta$. We find that $N = 1$ offers the best trade-off between performance and computational cost (discussion in Table 5).

**Reward hacking.** Reward hacking is a common issue in ReFL (Clark et al., 2023; Skalse et al., 2022) and also persists in face restoration tasks. It manifests as the restoration model generating adversarial samples to achieve higher reward scores, which lack diversity, exhibit uniformity, and contain unnatural artifacts, thus deviating from real face samples. To counteract this, we propose a strategy to dynamically update the Face Reward Model $\mathcal{R}$, concurrently with the training of the generator $g_\theta$. Specifically, after every $n$ training iterations of the generator $g_\theta$, we perform an update step for $\mathcal{R}$. In this update step, we utilize the most recent generator $g_\theta$ to produce a batch of high-quality restored images $\hat{\mathbf{I}}_{HQ}$. For each $\hat{\mathbf{I}}_{HQ}$, we have its corresponding original image $\mathbf{I}_{\text{HQ}}$ and the text description $\mathbf{T}$. Following the HPS v2 (Wu et al., 2023a), we employ $\mathcal{R}$ to compute similarity scores between the text description and each image: $s_{\text{HQ}} = \mathcal{R}(\mathbf{I}_{\text{HQ}}, \mathbf{T})$, $\hat{s}_{\text{HQ}} = \mathcal{R}(\hat{\mathbf{I}}_{\text{HQ}}, \mathbf{T})$. These pair scores are then converted into preference probabilities.

Let $\mathbf{I}_w = \mathbf{I}_{\text{HQ}}$ (the preferred, "winner" image) and $\mathbf{I}_l = \hat{\mathbf{I}}_{\text{HQ}}$ (the less preferred, "loser" image). The probability that $\mathbf{I}_w$ is preferred over $\mathbf{I}_l$ given the prompt $\mathbf{T}$ is formulated using a softmax-like function over their scores:

$$P(\mathbf{I}_w \succ \mathbf{I}_l|\mathbf{T}) = \frac{\exp(s_{\text{HQ}})}{\exp(s_{\text{HQ}}) + \exp(\hat{s}_{\text{HQ}})}. \tag{7}$$

To update the parameters of $\mathcal{R}$, we encourages this probability to be high, reflecting the fixed preference for $\mathbf{I}_{\text{HQ}}$ over $\hat{\mathbf{I}}_{\text{HQ}}$. Thus, we use a simplified version of entropy loss as our objective function:

$$\mathcal{L}_{\text{FRM}} = -\log P(\mathbf{I}_w \succ \mathbf{I}_l|\mathbf{T}). \tag{8}$$

By assigning a preference solely to $\mathbf{I}_{\text{HQ}}$, we ensure that the $\mathcal{R}$ is constrained to remain within the manifold space of real face images, thereby alleviating the occurrence of reward hacking driven by unstable rewards.

## 4 EXPERIMENTS

### 4.1 EXPERIMENTAL SETTINGS

We takes DiffBIR and OSEDiff as base and employ our proposed methods on them respectively. See Appendix B for implementation details.

**Training and Testing Data.** We used the FFHQ dataset (Karras et al., 2021) for training, which contains 70,000 high-quality facial images. During training, these images are resized to $512 \times 512$. Our strategy for synthesizing LQ faces from HQ ones during the training period is detailed in Appendix B. Follow the previous work (Wang et al., 2021; Gu et al., 2022), we employ the synthetic dataset CelebA-Test and two real-world datasets (*i.e.*, LFW-Test and WebPhoto-Test) to validate our proposed method.

**Evaluation Metrics.** On the Celeba-Test dataset, we used five reference metrics: SSIM (Wang et al., 2004), PSNR, LPIPS (Zhang et al., 2018), CLIP Score(Hessel et al., 2021), Deg. (Wang et al., 2021), and LMD (Gu et al., 2022), along with four non-reference metrics: MUSIQ (Ke et al., 2021), MANIQA (Yang et al., 2022) and FID (Heusel et al., 2017). To evaluate the aesthetic quality of generated face images on the CelebA-Test dataset, we utilized the LAION-AI aesthetic predictor to predict aesthetic scores, which are correlated with human preferences (LAION-AI, 2022). In addition, we used our pretrained FRM to score the restored face images, denoting as FaceReward.

**Comparison Methods.** We compare with not only the base models but also the latest state-of-the-art methods, including GFPGAN (Chan et al., 2021), CodeFormer (Zhou et al., 2022), VQFR (Gu et al., 2022), DR2+SPAR (Wang et al., 2023b), RestoreFormer (Wang et al., 2022), DifFace (Yue & Loy, 2024), OSEDiff (Wu et al., 2024), and DiffBIR (Lin et al., 2024).

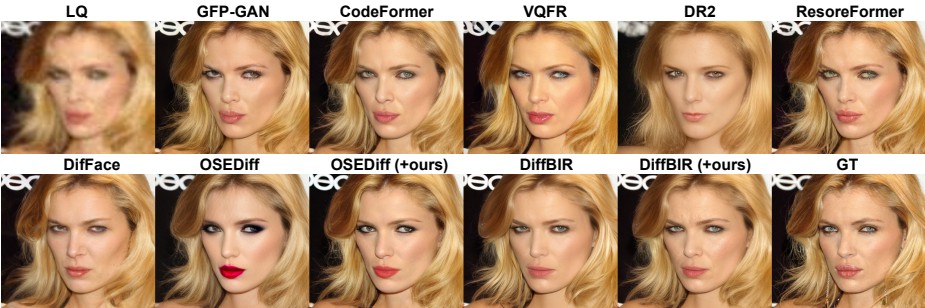

Figure 4: Qualitative comparison on the CelebA-Test. (Zoom in for details)

Table 1: Performance comparison of face restoration methods on CelebA-Test datasets. The highest score for each metric is highlighted in red, and the second-highest in blue. Metrics with ↑ indicate higher is better, while ↓ indicate lower is better. The values in parentheses represent our method's improvements over base models.

| Methods | SSIM↑ | PSNR↑ | LPIPS↓ | CLIP Score↑ | Deg.↓ | LMD↓ | MUSIQ↑ | MANIQA↑ | FID↓ | Aesthetic↑ | FaceReward↑ |
|---|---|---|---|---|---|---|---|---|---|---|---|
| Input | 0.6994 | 25.33 | 0.4866 | 0.7894 | 47.94 | 3.756 | 17.00 | 0.3957 | 143.95 | 4.0484 | 0.3397 |
| GFPGAN | 0.6772 | 24.65 | 0.3646 | 0.8410 | 34.58 | 2.4110 | 73.90 | 0.6522 | 42.57 | 5.6992 | 0.0741 |
| CodeFormer | 0.6925 | 25.85 | 0.3335 | 0.8931 | 31.08 | 1.9963 | 74.23 | 0.6520 | 45.57 | 5.8103 | 0.2864 |
| VQFR | 0.6654 | 23.76 | 0.3557 | 0.8562 | 42.48 | 2.9444 | 73.84 | 0.6544 | 46.77 | 5.7844 | 0.3142 |
| DR2+SPAR | 0.6512 | 22.89 | 0.4146 | 0.7437 | 57.24 | 4.5449 | 70.19 | 0.6374 | 62.54 | 5.6602 | 0.2455 |
| RestoreFormer | 0.6527 | 24.63 | 0.3652 | 0.8876 | 32.14 | 2.3020 | 73.75 | 0.6477 | 41.68 | 5.8015 | 0.2423 |
| DifFace | 0.6762 | 24.80 | 0.3994 | 0.8380 | 45.81 | 2.9766 | 68.96 | 0.6204 | 37.88 | 5.4708 | 0.3372 |
| OSEDiff | 0.6864 | 23.96 | 0.3478 | 0.7962 | 46.20 | 2.8871 | 73.41 | 0.6560 | 65.13 | 5.7720 | 0.2608 |
| OSEDiff (+ours) | 0.6838 | 24.93 | 0.3451 | 0.8732 | 38.41 | 2.4060 | 75.24 | 0.6640 | 44.40 | 5.9529 | 0.4389 |
|  | (-0.0026) | (+0.97) | (+0.0027) | (+0.0770) | (+7.79) | (+0.4811) | (+1.83) | (+0.0080) | (+20.73) | (+0.1809) | (+0.1781) |
| DiffBIR | 0.6775 | 25.44 | 0.3811 | 0.8877 | 35.16 | 2.2661 | 74.46 | 0.6752 | 45.50 | 5.7943 | 0.1938 |
| DiffBIR (+ours) | 0.7043 | 26.33 | 0.3454 | 0.9001 | 30.61 | 1.8642 | 74.82 | 0.6630 | 42.59 | 5.8475 | 0.4275 |
|  | (+0.0268) | (+0.89) | (+0.0357) | (+0.0124) | (+4.55) | (+0.4019) | (+0.36) | (-0.0122) | (+2.91) | (+0.0532) | (+0.2337) |

## 4.2 MAIN RESULTS

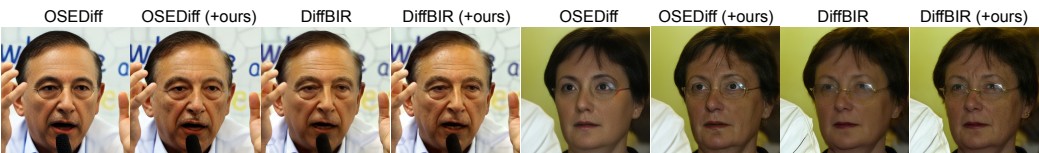

Figure 5: Qualitative comparison between the base model and the our methods on real-world faces.

**Evaluation on Synthetic Dataset.** We first show the quantitative comparison on the CelebA-Test in Table 1. We employed 11 metrics to comprehensively evaluate the overall performance of each method. Initially, a glance at the values within parentheses reveals that our approach achieves

performance improvements across nearly all metrics when compared to the base models. Comparing to state-of-the-art (SOTA) methods, the OSEDiff (+ours) and DiffBIR (+ours) achieve top rankings in the majority of metrics, such as Deg., LMD, Aesthetic, and FaceReward, indicating that our proposed ReFL framework can enhance perceived face quality while preserving identity consistency. As the shown qualitative comparisons in Figure 4, our method exhibits superior identity consistency and skin texture details.

**Evaluation on Real-world Datasets.** Table 2 shows the quantitative results. We find that our proposed ReFL framework improves the aesthetic score and MUSIQ, which measures image quality. Comparing to other methods, OSEDiff (+ours) achieves the best performance on both datasets From the qualitative results in Figure 5, a qualitative comparison between the base model and ReFL is presented. We observe that the base models, when faced with real-world degradation, often fails to restore facial details, resulting in a smooth face. Our method overcomes these problems and generate realistic faces with richer details.

Table 2: Performance comparison of face restoration methods on wild datasets. The highest score for each metric is highlighted in red, and the second-highest in blue. Metrics with ↑ indicate higher is better. The values in parentheses represent our method's improvements over base models.

| Dataset | LFW-Test | | WebPhoto | |
|---|---|---|---|---|
| Methods | Aesthetic↑ | MUSIQ↑ | Aesthetic↑ | MUSIQ↑ |
| Input | 4.9978 | 26.87 | 4.2584 | 18.63 |
| GFP-GAN | 5.6042 | 73.57 | 5.2473 | 72.09 |
| CodeFormer | 5.6414 | 70.69 | 5.1860 | 71.16 |
| VQFR | 5.6802 | 74.39 | 5.2829 | 70.91 |
| DR2+SPAR | 5.5409 | 72.22 | 5.1785 | 63.65 |
| RestoreFormer | 5.6068 | 73.70 | 5.1213 | 69.84 |
| DiffFace | 5.4104 | 69.85 | 5.0721 | 65.21 |
| OSEDiff | 5.6796 | 73.40 | 5.4161 | 72.60 |
| OSEDiff (+ours) | 5.7183 (+0.0387) | 74.81 (+1.41) | 5.5412 (+0.1251) | 74.05 (+1.45) |
| DiffBIR | 5.6814 | 73.71 | 5.2728 | 67.45 |
| DiffBIR (+ours) | 5.6860 (+0.0046) | 74.49 (+0.78) | 5.3554 (+0.0826) | 71.38 (+3.93) |

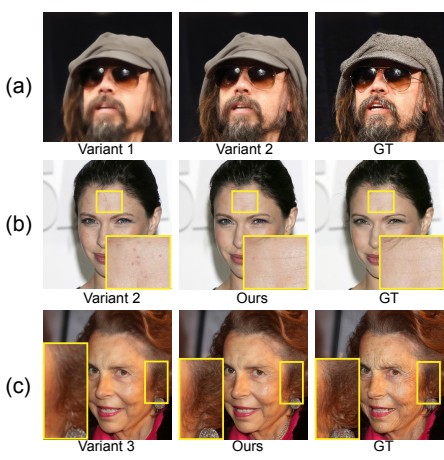

Figure 6: Ablation study visualizations.

Table 3: Performance comparison of FRM and HPS v2 reward models

| Reward Type | MANIQA↓ | MUSIQ↑ | FID↓ |
|---|---|---|---|
| HPS v2 | 0.6630 | 69.78 | 48.94 |
| FRM (ours) | **0.6535** | **74.82** | **42.59** |

Table 4: Ablation study of ReFL components

| Struct | SC | WR | Rwd | RU | LMD↓ | MUSIQ↑ | Aesthetic↑ |
|---|---|---|---|---|---|---|---|
| Base | | | | | 2.2661 | 74.46 | 5.7943 |
| Variant 1 | ✓ | ✓ | | | 1.9583 | 54.70 | 5.6572 |
| Variant 2 | ✓ | ✓ | ✓ | | 1.8834 | 71.12 | 5.6063 |
| Variant 3 | ✓ | | ✓ | ✓ | 1.8644 | 70.67 | 5.7528 |
| Ours | ✓ | ✓ | ✓ | ✓ | 1.8642 | 74.82 | 5.8475 |

## 4.3 ABLATION STUDY

We conduct main ablation study based on DiffBIR (+ours) on CelebA-Test dataset, and the ablation study based on OSEDiff (+ours) is provided in Appendix D.1. First, we manually annotate 360 pairs of face images and calculate the preference prediction accuracy of HPS v2 and our FRM. Our FRM outperforms HPS v2 significantly (*i.e.*, 87.78% v.s. 63.05%), demonstrating a high alignment with human preferences and superior human perception. Furthermore, when we replace our FRM with the original HPS v2 model for the ReFL framework and keep the same training configurations, our FRM obviously outperfoms HPS v2, as shown in Table 3.

Second, we decompose our proposed ReFL framework into four components, including structural consistency constraint (SC), weight regularization constraint (WR), using reward feedback (Rwd), and updating reward model (RU), resulting in three variants. As shown in Table 4, Variant 1 (employing SC and WR without FRM components) improves identity preservation (LMD) but degrades perceptual quality (MUSIQ), resulting in overly smooth faces (See Figure 6(a)). After adding Rwd to Variant 1, obtaining Variant 2, we find obvious enhancements in perceptual quality (MUSIQ) and restores finer facial details (See Figure 6(a) and Table 4).

Removing WR from ours entire ReFL framework (*i.e.*, Variant 3) leads to a decline in perceptual quality, identity consistency, and aesthetic scores (See Table 4). This is attributed to the disruption of

pre-trained priors and weakened generation capabilities, as evidenced by the loss of hair details in Variant 3 (See Figure 6(b)). Finally, we validate that the dynamic update mechanism of FRM (RU) is crucial for the reward hacking. In Figure 6(c), Variant 2 exhibits "reward hacking", generating faces with stereotypical artifacts like acne marks. Incorporating RU eliminates these artifacts, improving generation quality and outperforming Variant 2, as shown in Table 4.

To manage the computational cost of fine-tuning our ReFL-based restoration model, we employ truncated backpropagation for the final $N$ denoising steps. We evaluated $N \in \{1, 5, 20\}$ in Table 5 and observed that while larger $N$ yields marginal gains in distortion metrics (e.g., SSIM), it notably degrades key perceptual metrics (FID, FaceReward) and increases training overhead. Consequently, we adopt $N = 1$ in all our experiments to achieve the best trade-off between restoration quality and training efficiency. More ablation experiments are provided in Appendix D.

Table 5: Performance under different backpropagation truncation steps ($N$).

| Steps ($N$) | SSIM↑ | PSNR↑ | LPIPS↓ | CLIP↑ | Deg.↓ | LMD↓ | MUSIQ↑ | MANIQA↑ | FID↓ | Aesthetic↑ | FaceRwd↑ |
|---|---|---|---|---|---|---|---|---|---|---|---|
| 1 | 0.7043 | 26.33 | 0.3454 | 0.9001 | 30.61 | 1.8642 | 74.82 | 0.6630 | 42.59 | 5.8475 | 0.4275 |
| 5 | 0.7101 | 26.42 | 0.3221 | 0.9103 | 30.10 | 1.8013 | 73.68 | 0.6652 | 47.27 | 6.0627 | 0.3876 |
| 20 | 0.7151 | 26.37 | 0.3382 | 0.9073 | 30.11 | 1.8031 | 73.60 | 0.6630 | 46.32 | 6.0751 | 0.3923 |

**Analysis of Training Dynamics.** As shown in Figure 7, the training of DiffBIR (+Ours) converges stably. The Face Reward Score rises quickly at the beginning and then plateaus, with a slight decrease in the later stage due to our dynamic reward update. This mechanism deliberately tightens the reward, guiding the model back toward the real-face manifold and suppressing "reward hacking" (i.e., chasing scores at the cost of realism), which is consistent with the theoretical dynamics in Figure 12(right). Meanwhile, the convergence of the structural consistency loss signifies an improvement in facial identity preservation.

**User Study.** We further conducted a pairwise user study comparing our results (+Ours) with the corresponding baselines (OSEDiff, DiffBIR) in terms of identity fidelity and visual realism. As shown in Table 6 reveals that the faces restored by our approach are overwhelmingly preferred by human subjects. The standards for our user study are detailed in Appendix C.

Table 6: Human preference ratio between our method and the base Model for realism and fidelity.

| Comparison | Fidelity | Realism |
|---|---|---|
| OSEDiff (+ours) vs OSEDiff | 78% vs 22% | 88% vs 12% |
| DiffBIR (+ours) vs DiffBIR | 68% vs 32% | 75% vs 25% |

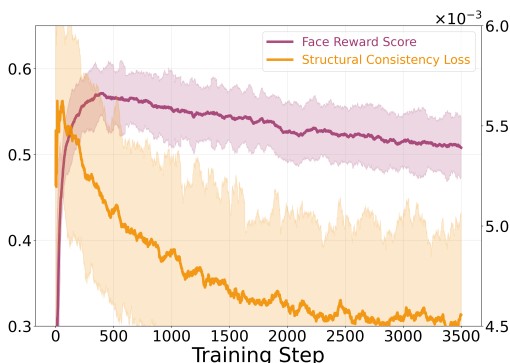

Figure 7: Reward and loss curves during training.

## 5 CONCLUSION

To tackle key challenges in diffusion-based face restoration–such as insufficient facial detail and poor identity preservation–we introduce *DiffusionReward*, a method that fine-tunes the denoising process via the ReFL framework. In the ReFL framework, we not only show a data curation pipeline for buiding a powerful FRM but also propose useful constraints for optimizing the diffusion denoising process. Moreover, we propose a dynamic updating stategy to avert the reward hacking problem.

## 6 LIMITATION

DiffusionReward framework has been primarily validated on diffusion-based face restoration methods (e.g., DiffBIR and OSEDiff). Its core ReFL mechanism, particularly the integration of gradient flow and the dynamic updates to the FRM, was designed considering the characteristics of diffusion models. Consequently, ReFL yields limited performance gains for certain non-diffusion methods, such as GFPGAN and CodeFormer. The quantitative experimental results are presented in Table 15 of Appendix G.1. We attribute this observation to the inherent lack of stochasticity during the generation process of these architectures, which limits the exploration needed by the face reward model.

ETHICS STATEMENT

All authors adhere to the ICLR Code of Ethics. Our research focuses solely on the technical challenge of image restoration and does not introduce new ethical concerns. All experiments were performed using publicly available datasets for both training and evaluation.

REPRODUCIBILITY STATEMENT

To ensure reproducibility, we will publicly release our source code and models. Our experiments are conducted using public datasets, and all implementation details, training hyperparameters are provided in the Appendix.

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

# Appendix

## Table of Contents

# A  IMPLEMENTATION DETAILS OF FACE REWARD MODEL

This section provides supplementary details to those presented in Sec. 3.1.

## A.1  DETAILS OF TRAINING DATA ANNOTATION

To effectively train our Face Reward Model (FRM), it is crucial to prepare accurate textual descriptions and preference labels for the face images.

**Text Description Generation for Face Images.** High-quality textual descriptions enable the reward model to better comprehend image content, thereby providing more precise feedback. Our FRM training data originates from a public face dataset (Wu et al., 2023b) containing 19,590 face images. For these images, we generated corresponding textual descriptions as follows: We utilized the LLaVA (Liu et al., 2023) model to automatically generate text descriptions for each facial image. When inputting an image to the LLaVA model, we employed the following carefully designed prompt:

Listing 1: Prompt for LLaVA model

```
As an AI face caption expert, please provide precise description for
    face.
Provide a simple description of the face, including gender, age, facial
features, pose (whether the person is in profile, front-facing, looking
    up,
etc.), and facial expression. Begin your description with 'The face'.
If the image includes one or more elements from list [HAIR, BEARD,
    CLOTHES,
GLASSES, HEADWEAR, FACEWEAR, JEWELRY, FACE PAINT, HAND, HANDHELD ITEMS],
include descriptions of those elements. (Word limit: within 35 words.)
```

The primary objective of this prompt was to ensure that the generated text descriptions not only cover fundamental facial attributes (such as gender, age, facial features, and expression) but also specifically emphasize the person's pose (e.g., profile, front-facing, looking up) and any potential occlusions or adornments (such as hair, beard, clothes, glasses, headwear, facewear, jewelry, face paint, hands, or handheld items). By doing so, we aimed for the text descriptions to guide the reward model towards a more comprehensive and detailed perception of the image, thereby enhancing the accuracy of the reward scores. Similarly, during the training process of DiffusionReward, we added text descriptions to the training dataset FFHQ (Karras et al., 2021). In Figure 8, we present the face images along with their corresponding text descriptions.

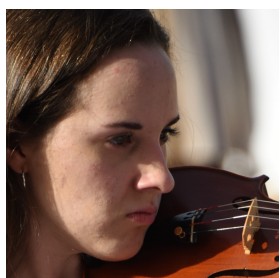 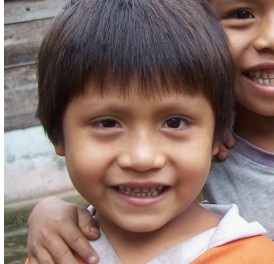 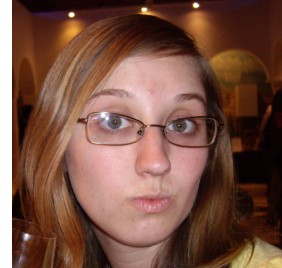

*The face of a young woman with fair skin and light brown hair, wearing a serious expression, holding a violin.*

*The face of a young boy with short black hair, brown eyes, and a wide smile, wearing an orange shirt. The background shows another smiling child and a wooden structure.*

*The face of a young woman with light skin and straight, shoulder-length blonde hair, wearing glasses and a yellow top. She is front-facing, making a kissing face, with a background of a dimly lit room and indistinct figures.*

Figure 8: Text description example

**Manual Annotation of Preference Labels.** To acquire reliable human preference data, we organized a team of three annotators to manually label image pairs. In total, the annotators provided preference selections for 3,600 image pairs. We established clear annotation guidelines for the human annotators to ensure consistency and quality:

When presented with two facial images generated by different face restoration models, annotators were instructed to select the image they preferred. This preference decision was based on a comprehensive consideration of the following three core rules, ordered by importance:

- *Realism of the Facial Image:* This was the most critical factor. Annotators were required to meticulously inspect the images for any unnatural artifacts, distortions, blurring, or other signs of unnaturalness. The image should appear as close as possible to a real-world photograph of a face.
- *Richness and Naturalness of Facial Details:* Annotators assessed whether the facial details (such as skin texture, hair, and clarity of facial features) were sufficiently rich and whether these details conformed to the natural texture characteristics of a real face, avoiding overly smooth details.
- *Consistency between the Facial Image and its Textual Description:* This was the final consideration. Annotators needed to judge if the image content aligned with the text description.

*The face of a middle-aged man with a dark beard, wearing a gray Civil War-era hat with a black brim.*

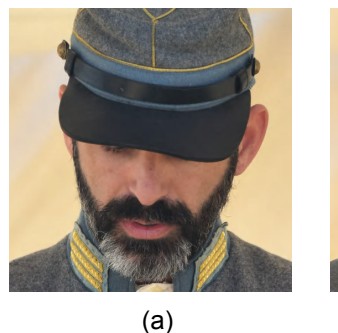 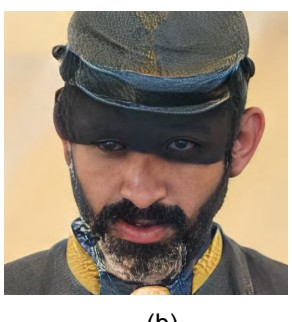

(a)          (b)

Figure 9: The brim and eyes of (b) have artifacts, so (a) is a better face image.

*The face of a smiling woman with long, wavy brown hair, light skin, and red lipstick.*

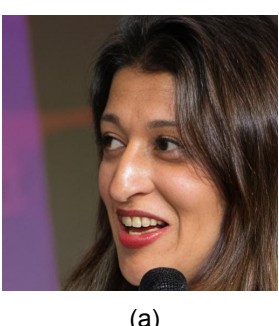 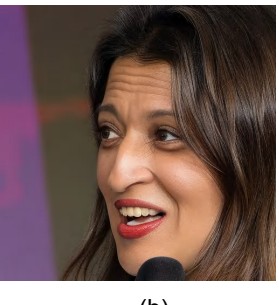

(a)          (b)

Figure 10: Sample (a) exhibits more realistic textures, rendering it the superior choice.

The final preference judgment was based on a holistic assessment considering these three rules. To further illustrate the application of this hierarchical decision-making process, annotators proceeded as follows:

First, they evaluated the images for any obvious, unrealistic artifacts based on the primary rule of realism. For instance, as demonstrated in Figure 9, if image (b) exhibited distorted elements such as a warped cap brim or unnatural-looking eyes when compared to image (a), Figure 9 (a) would be selected as the superior image. If both images passed the initial realism check, the focus shifted to the second rule: the richness and naturalness of facial details. As exemplified in Figure 10, if the skin in image (b) appeared overly smooth and artificial, while image (a) preserved fine and natural skin textures, then Figure 10 (a) would be deemed the better facial image.Finally, if a clear preference could not be established based on the first two rules, the third rule concerning text-image consistency

was applied. For example, as depicted in Figure 11, if image (b) was missing an element explicitly mentioned in its textual description, such as 'glasses', whereas Figure 11 (a) accurately reflected the description, then Figure 11 (a) would be chosen as the preferred image.

Through this structured process, we aimed to collect preference data that accurately reflects human subjective perception of image quality, grounded in both the objective visual content and the semantic information conveyed by the textual descriptions.

**Automated Annotation Pipeline.** To scale up the collection of preference labels beyond the 3,600 manually annotated pairs and efficiently construct a large dataset for training our FRM, we developed an automated annotation pipeline. This pipeline leverages a Support Vector Machine (SVM) Cortes & Vapnik (1995) classifier trained on the previously described human-annotated data.

The face of a middle-aged man with a beard, glasses, and an open-mouthed expression, bathed in red light.

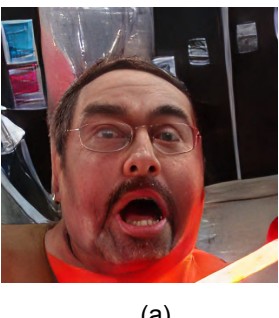 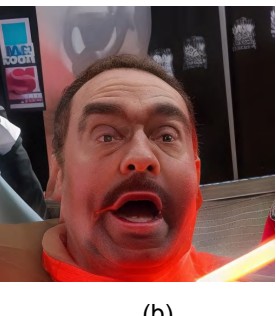

(a)           (b)

Figure 11: Sample (a) successfully restored the glasses mentioned in the text description. Therefore, (a) is the superior choice.

The 12-dimensional feature vectors $v$ (formed by concatenating the 6 evaluation metrics from each image in a pair, as detailed in Sec. 3.1 of the main paper and illustrated in Figure 2 therein) and the corresponding integer preference labels derived from the 3,600 human-annotated image pairs serve as the training set for this SVM classifier.

The SVM classifier was implemented using the `scikit-learn` library. The training process began with loading these feature vectors and labels. To enhance the SVM's performance and training stability, the feature vectors underwent standardization using a `StandardScaler`, which was fitted to the training data and then applied to transform it, ensuring each feature dimension had a mean of 0 and a variance of 1.

A Support Vector Classifier (`SVC`) was selected as the preference prediction model. To determine the optimal model configuration, we utilized `GridSearchCV` with 5-fold cross-validation on the training set. The hyperparameter search space included various kernel types (e.g., `'linear'`, `'rbf'`, `'poly'`), the regularization parameter `C`, and other kernel-specific parameters (such as `gamma` and `degree`). The grid search aimed to maximize the average cross-validation `accuracy`. Upon completion of the grid search, the best hyperparameter combination was identified. The trained `StandardScaler` and the optimized `SVC` model were then saved to disk for subsequent use.

Once trained, the SVM classifier was used to automatically assign preference labels to the remaining image pairs in our dataset that were not manually annotated. The procedure is as follows:

- For an unlabeled image pair, its 12-dimensional raw metric vector is extracted.
- The saved `StandardScaler` is applied to standardize this vector.
- The standardized feature vector is then fed into the trained SVM model.
- The SVM model outputs a predicted preference label (e.g., '1' indicating the first image is of higher quality, '0' indicating the second is better).

This hybrid approach, combining manual annotations with an efficient SVM-based automated pipeline, allowed us to effectively augment the dataset with a large number of preference labels. This provided a richer source of supervision for training the FRM while significantly reducing the cost and time associated with purely manual annotation.

A.2 THE TRAINING DETAILS OF FACE REWARD MODEL

The Face Reward Model (FRM) is a critical component of our DiffusionReward framework, designed to provide feedback signals that align the output of face restoration models with human preferences. Its training involves specific architectural choices, initialization, optimization parameters, and a tailored loss function.

The FRM utilizes the ViT-H-14 CLIP (Radford et al., 2021) architecture as its backbone. We initialize the model with pre-trained weights from HPS v2 (Wu et al., 2023a)[1]. CLIP consists of an image encoder $E_I$ and a text encoder $E_t$.

The FRM is fine-tuned on our curated face preference dataset . The training process employs the Adam optimizer. We fine-tune the model for 20,000 iterations with a learning rate of $3.3 \times 10^{-6}$. During fine-tuning, only specific parts of the model are made trainable to preserve the rich priors from pre-training while adapting to our specific task. Specifically, the last 20 layers of the image encoder ($E_I$) and the last 11 layers of the text encoder ($E_t$) are trainable. All other parameters are kept frozen.

The FRM is trained using pairwise preference data. Each training instance consists of a pair of images, denoted as $\{\mathbf{I}_1, \mathbf{I}_2\}$, a corresponding textual description $\mathbf{T}$, and a human preference label $y$. The label $y$ is typically a one-hot vector; for instance, $y = [1, 0]$ if image $\mathbf{I}_1$ is preferred over $\mathbf{I}_2$, and $y = [0, 1]$ otherwise.

The FRM computes a score for each image with respect to the text description. Let $\boldsymbol{e}_{i_1} = E_I(\mathbf{I}_1)$ and $\boldsymbol{e}_{i_2} = E_I(\mathbf{I}_2)$ be the image embeddings obtained from the image encoder $E_I$, and $\boldsymbol{e}_t = E_t(\mathbf{T})$ be the text embedding from the text encoder $E_t$. Following the principles of CLIP and HPS v2, and aligning with our notation in Sec. 3.1 of main paper, the preference scores $s_1$ and $s_2$ are derived from the cosine similarities:

$$s_k = \frac{\boldsymbol{e}_{i_k} \cdot \boldsymbol{e}_t}{\tau}$$

where $k \in \{1, 2\}$, $\theta$ represents the trainable parameters of the FRM, and $\tau$ is a learned temperature scalar inherent to the CLIP model, which scales the logits.

Given these scores for the pair of images, the predicted preference probability for image $\mathbf{I}_k$ (i.e., $\hat{y}_k$) is calculated using a softmax function, consistent with $\sigma([s_1; s_2])$ in Figure 2 of the main paper:

$$\hat{y}_k = \frac{\exp(s_k)}{\sum_{j=1}^{2} \exp(s_j)}$$

This results in a probability distribution $\hat{y} = [\hat{y}_1, \hat{y}_2]$ over the two images.

The parameters $\theta$ of the FRM are optimized by minimizing the cross-entropy loss ($\mathcal{L}_{\text{CE}}$ as denoted in Sec. 3.1 of main paper) between the ground-truth preference label $y = [y_1, y_2]$ and the predicted preference distribution $\hat{y} = [\hat{y}_1, \hat{y}_2]$. The $\mathcal{L}_{\text{CE}}$ Can be formulated as:

$$\mathcal{L}_{\text{CE}} = -\sum_{j=1}^{2} y_j \log(\hat{y}_j)$$

B THE IMPLEMENTATION DETAILS OF DIFFUSIONREWARD

This section is used to supplement the implementation details of Sec. 4 in the main paper.

Our strategy for synthesizing LQ faces from HQ ones during the training period is as follows:

$$\mathbf{I}_{\text{LQ}} = \left\{ \left[ (\mathbf{I}_{\text{HQ}} \otimes \boldsymbol{k}_\sigma)_{\downarrow_r} + \boldsymbol{n}_\delta \right]_{\text{JPEG}_q} \right\}_{\uparrow_r} \tag{9}$$

Where the HQ images are first convolved with a Gaussian kernel $\boldsymbol{k}_\sigma$, followed by a downsampling with a factor of $r$, and then corrupted with Gaussian noise $\boldsymbol{n}_\delta$. Subsequently, the images undergo JPEG compression with a quality factor of $q$. Finally, the LQ image is resized back to the original

---

[1]Source weights for HPS v2 are available at https://github.com/tgxs002/HPSv2.

$512 \times 512$. Here, $\sigma$, $r$, $\delta$, and $q$ are randomly sampled from the intervals $[0.1, 12]$, $[1, 12]$, $[0, 15]$, and $[30, 100]$, respectively.

Our DiffusionReward framework is developed by fine-tuning two pre-trained base models: DiffBIR-v1[2] and OSEDiff[3]. Both of these base models were originally pre-trained on the FFHQ face dataset. We initialize our training using their respective released pre-trained weights (e.g., the DiffBIR v1 Face version and the OSEDiff Face version). Subsequently, we apply our proposed Reward Feedback Learning (ReFL) strategy to further fine-tune these pre-trained models, resulting in two distinct versions of our DiffusionReward.

The denoising process within our DiffusionReward framework employs DDIM (Song et al., 2020) sampling. During the ReFL fine-tuning phase, distinct components were trained depending on the base model: for DiffBIR, we focused on training its ControlNet module, whereas for OSEDiff, we trained the LoRA parameters of its UNet.

The general training configuration utilized the Adam optimizer with a learning rate of $5 \times 10^{-5}$ and a batch size of 12. All training was conducted on an NVIDIA L20 GPU equiped with 48GB of memory. For the ReFL training specifically with OSEDiff as the base, the loss weighting hyperparameters were set as follows: $\lambda_{\text{LPIPS}} = 0.02$, $\lambda_{\text{DWT}} = 0.01$, $\lambda_{\text{reward}} = 0.005$, and $\lambda_{\text{reg}} = 1$. When DiffBIR served as the base model for ReFL training, the corresponding hyperparameters were: $\lambda_{\text{LPIPS}} = 0.01$, $\lambda_{\text{DWT}} = 0.01$, $\lambda_{\text{reward}} = 0.005$, and $\lambda_{\text{reg}} = 10^{-4}$. Furthermore, a crucial aspect of our ReFL training strategy involved the dynamic update of the Face Reward Model ($\mathcal{R}$); this update was performed every $n = 10$ training iterations of the main restoration model.

## C  USER STUDY

We conducted a user study by randomly selecting 100 face images from the CelebA test dataset. We invited 20 participants with different backgrounds to perform a pairwise comparison between the results generated by our method (+ours) and the corresponding baseline models (OSEDiff, DiffBIR). Participants were asked to choose their preferred result based on two core criteria:

1. **Fidelity:** Which image better preserves the identity features of the original person?

2. **Realism:** Which image looks more natural and realistic, with fewer artifacts?

The statistical results of the study are presented in Table 7, which shows a clear preference for the results enhanced by our method.

Table 7: Human preference evaluation

| Comparison | Fidelity Preference % | Realism Preference % |
|---|---|---|
| OSEDiff (+ours) vs OSEDiff | 78% vs 22% | 88% vs 12% |
| DiffBIR (+ours) vs DiffBIR | 68% vs 32% | 75% vs 25% |

## D  MORE ABLATION ANALYSIS

### D.1  ABLATION OF OSEDIFF

In Sec. 4.3 of the main paper, due to space constraints, we presented ablation studies primarily for the DiffusionReward framework applied to DiffBIR. Here, we provide additional ablation results specifically for DiffusionReward when OSEDiff is used as the base model. These results are summarized in Table 8. The conclusions in the table are consistent with the analysis previously conducted in Sec. 4. The structural consistency constraint (SC), weight regularization constraint (WR), reward feedback (Rwd), and updating reward model (RU) work together to improve the quality of face restoration.

---

[2]Source weights for DiffBIR are available at https://github.com/XPixelGroup/DiffBIR.
[3]Source weights for OSEDiff are available at https://github.com/cswry/OSEDiff.

Table 8: Ablation Study of ReFL Components

| Struct | SC | WR | Rwd | RU | LMD↓ | MUSIQ↑ | Aesthetic↑ |
|--------|----|----|----|----|------|--------|-----------|
| Base | | | | | 2.8871 | 73.41 | 5.7720 |
| Variant 1 | ✓ | ✓ | | | 2.3406 | 69.85 | 5.7813 |
| Variant 2 | ✓ | ✓ | ✓ | | 2.3997 | 69.97 | 5.8912 |
| Variant 3 | ✓ | | ✓ | ✓ | 2.3962 | 70.83 | 5.7860 |
| DiffusionReward (OSEDiff) | ✓ | ✓ | ✓ | ✓ | 2.4060 | 75.24 | 5.9529 |

## D.2 PREFERENCE PREDICTOR ARCHITECTURE SELECTION

When designing our preference predictor, a key goal was to simplify the modeling of human preference from image data. Instead of using high-dimensional pixels, we first extract a set of established proxy metrics (such as SSIM, NIQE, LPIPS, etc.) known to correlate with human-perceived quality. These metrics form a low-dimensional feature vector for each image.

For this relatively simple, low-dimensional feature space, we hypothesized that a traditional and robust classifier like a SVM might generalize better and be less prone to overfitting than a deep learning model with a larger number of parameters. To validate this hypothesis, we conducted a direct comparative experiment between an SVM and MLPs with different depths. We trained each model as a preference predictor and evaluated its accuracy on our manually annotated preference dataset. The results of this comparison are summarized in Table 9.

Table 9: Prediction accuracy on our human-annotated dataset for different predictor architectures.

| Predictor Architecture | Prediction Accuracy |
|-----------------------|---------------------|
| MLP (3-layer) | 69.2% |
| MLP (4-layer) | 68.8% |
| SVM | **70.0%** |

As the table shows, the SVM classifier achieved the highest prediction accuracy in our task setting. This result supports our choice of SVM, which is based on direct experimental evidence. This predictor provided reliable data annotation for the subsequent training of our high-quality FRM.

## D.3 STABILITY ANALYSIS OF THE FRM DURING DYNAMIC UPDATES

A critical challenge in training stage is ensuring the reward model remains reliable and aligned with human preferences throughout the dynamic updating process. A deteriorating reward model could lead to sparse gradients or optimization collapse. To investigate this, we tracked the "Human Consistency" of our FRM on the manually annotated test set (360 pairs) at intervals of 500 training iterations.

As shown in Table 10, the FRM exhibits stable human consistency during the training stage. The alignment starts at 87.78% and remains above 83.06% even after 3,000 iterations. This consistency is significantly higher than that of the baseline HPSv2 (approximately 63%), ensuring that the reward signal stays dense and reliable during training stage.

We observe a slight gradual decline in consistency (from 87.78% to 83.06%). This phenomenon is attributed to the manifold alignment objective in our dynamic update strategy. As discussed in Section 3.3, the dynamic update minimizes the probability of generated samples being preferred over real images (i.e. Eq. 8). This mechanism strictly constrains the reward model to align with the real face distribution, penalizing any deviations from the real manifold to prevent the restoration model from exploiting loopholes in the reward function (i.e., reward hacking). Consequently, this stringent focus on realism incurs a necessary trade-off: a minor loss in human consistency in exchange for robust defense against reward hacking, thereby ensuring the generation of photorealistic results.

Table 10: Human Consistency of the Face Reward Model (FRM) during Dynamic Training.

| Iteration | 0 | 500 | 1000 | 1500 | 2000 | 2500 | 3000 |
|---|---|---|---|---|---|---|---|
| Human Consistency (%) | 87.78 | 86.11 | 85.83 | 85.00 | 83.89 | 83.01 | 83.06 |

## D.4 ABLATION STUDY ON THE ROLE OF TEXT INPUT IN THE REWARD MODEL

Our FRM is designed to generate a scalar reward score by evaluating a pair of inputs: the restored image and a corresponding text description. This reward is then used to compute a Reward Loss, which guides the optimization of the restoration network. This design aims to align the model's output with human preference.

The role of each input modality is distinct. The restored image allows the FRM to assess holistic qualities such as realism, detail richness, and overall aesthetic appeal, providing a perception-aligned learning signal. The text description, in turn, acts as a semantic anchor. It provides essential context (e.g., facial features, age, accessories) that enables the FRM to evaluate not only the visual quality of the restoration but also its semantic plausibility. This ensures that generated details are contextually appropriate, rather than being arbitrary high-frequency textures.

To empirically validate the contribution of the text description, we conducted an ablation study. We trained two versions of the FRM: one utilizing the full "Image & Text" input and a control version using only the image with a null text input. We then measured how accurately each model's predictions aligned with true human preferences on a manually annotated test set of 360 image pairs.

The results, presented in Table 11, demonstrate the effectiveness of incorporating semantic context.

Table 11: Ablation study on the impact of text descriptions.

| FRM Input Methods | Human Consistency (↑) |
|---|---|
| Image & Text | 87.78% |
| Image & Null Text | 85.01% |

The data clearly indicates that while a model trained on images alone is effective, the inclusion of text descriptions allows the FRM's judgments to align more closely with human preferences (accuracy increased from 85.01% to 87.78%). This confirms that the text provides a more precise and semantically grounded reward signal, which is crucial for guiding the restoration process.

## D.5 SENSITIVITY ANALYSIS ON THE SCALE OF HUMAN ANNOTATION

To address the concern regarding the sensitivity of our framework to the amount of human-annotated data, we conducted an ablation study by varying the proportion of manual annotations used to train the SVM preference predictor. Specifically, we evaluated three settings: 0% (relying solely on the pre-trained HPSv2 without domain-specific fine-tuning), 50% (using half of the manual annotations), and 100% (our full setting). We assessed both the alignment consistency of the face reward model with human judgments and the final restoration quality of the DiffBIR (+ours) trained with these respective reward signals.

The quantitative results are summarized in Table 12. We observe a clear positive correlation between the scale of human annotation and the model performance:

• Using 0% annotation (i.e., raw HPSv2), the alignment with human preference on our face dataset is relatively low (69.78%). Incorporating just 50% of the manual data significantly boosts this alignment to 83.21%, and utilizing 100% of the data further elevates it to 87.78%. This demonstrates that domain-specific human feedback is crucial for calibrating the reward model to the nuances of face restoration.

• The improvement in the reward model directly translates to better restoration outcomes. As the annotation ratio increases, the perceptual quality metric (MUSIQ) improves from 69.78 to 74.82, and the distributional distance to real images (FID) decreases significantly from 48.94 to 42.59.

These results indicate that while the base HPSv2 provides a foundational perception of quality, our manual annotation process effectively bridges the domain gap, enabling the restoration model to generate more realistic and human-preferred facial details.

Table 12: Impact of human annotation scale on reward model alignment and restoration quality.

| Annotation Ratio | Human Consistency (↑) | MANIQA (↑) | MUSIQ (↑) | FID (↓) |
|---|---|---|---|---|
| 0% | 69.78% | 0.6630 | 69.78 | 48.94 |
| 50% | 83.21% | 0.6689 | 73.32 | 45.90 |
| 100% (ours) | 87.78% | 0.6535 | 74.82 | 42.59 |

### D.6 SENSITIVITY TO WEIGHT REGULARIZATION STRENGTH

To decouple the contributions of the Weight Regularization (WR) constraint and the Face Reward Model (FRM), we conducted a sensitivity analysis on the WR hyper-parameter $\lambda_{reg}$. This analysis aims to rigorously verify whether the observed perceptual improvements stem from the reward guidance or merely from optimal tuning of the regularization weight.

We evaluated both Variant 1 (equipped only with SC and WR losses, excluding FRM) and our proposed method (Ours) across three orders of magnitude for $\lambda_{reg}$: $\{10^{-3}, 10^{-4}, 10^{-5}\}$. The quantitative results, summarized in Table 13, reveal distinct behaviors. As shown in the upper section of Table 13, simply adjusting $\lambda_{reg}$ in Variant 1 fails to yield significant perceptual improvements. Regardless of the regularization strength, MUSIQ scores remain plateaued in the range of 54–57, and Aesthetic scores hover around 5.6–5.7. This confirms that the WR loss serves primarily to maintain the original generative capability (i.e., acting as an anchor) rather than driving perceptual enhancement.

In contrast, incorporating the FRM triggers a substantial performance leap. Even with a suboptimal $\lambda_{reg}$ of $10^{-3}$, our method achieves a MUSIQ score of 69.50, far surpassing the best result of Variant 1 (57.57). With the optimal $\lambda_{reg} = 10^{-4}$, our method peaks at a MUSIQ score of 74.82 with superior identity preservation. This empirically proves that the FRM is indispensable for achieving high-quality restoration.

Table 13: Sensitivity analysis of the Weight Regularization hyper-parameter ($\lambda_{reg}$). We compare Variant 1 (w/o FRM) and Ours (w/ FRM) under different regularization weights.

| Method | $\lambda_{reg}$ | LMD (↓) | MUSIQ (↑) | Aesthetics (↑) |
|---|---|---|---|---|
| *Variant 1* | $10^{-3}$ | 2.0252 | 57.57 | 5.7358 |
| | $10^{-4}$ | 1.9583 | 54.70 | 5.6572 |
| | $10^{-5}$ | 1.9087 | 55.05 | 5.7729 |
| *Ours* | $10^{-3}$ | 1.9467 | 69.50 | 5.8117 |
| | $10^{-4}$ | 1.8642 | 74.82 | 5.8475 |
| | $10^{-5}$ | 1.8182 | 73.22 | 5.8231 |

### E ANALYSIS OF TRAINING COSTS AND INFERENCE EFFICIENCY

In this section, we provide a detailed quantitative analysis of the training overhead and inference latency of the proposed DiffusionReward. Our objective is to demonstrate that our method achieves significant improvements in image restoration quality without introducing any additional inference burden, while keeping training costs within a highly reasonable range.

We quantitatively analyze DiffusionReward from the perspectives of both training overhead and inference cost. It is crucial to emphasize that the auxiliary modules introduced in our framework—including the Face Reward Model are utilized exclusively during the training stage. Once training is finalized, all auxiliary networks and components are discarded, enabling efficient inference.

During inference stage, the network architecture remains identical to the base model. Consequently, the inference speed of our method is inherently determined by the chosen base model. As shown in Table 14, our approach maintains the exact same inference latency as the respective baselines. Notably, when applied to efficient one-step sampling methods like OSEDiff, our framework fully preserves its rapid inference capability, ensuring seamless integration into existing pipelines without compromising real-time processing.

Regarding training overhead, our method performs post-training refinement on an off-the-shelf restoration model, requiring only minimal fine-tuning to achieve improved restoration quality. Meanwhile, while backpropagating gradients through the image decoder and reward model is theoretically expensive, our adoption of truncated backpropagation (with $N = 1$) effectively circumvents memory bottlenecks and computational prohibitive costs. As evidenced by the training-performance ratio in Table 14, this strategy yields an exceptionally high return on investment. Compared to the substantial cost of pre-training from scratch, our fine-tuning approach incurs only marginal additional overhead: for OSEDiff, it requires only 24 additional GPU hours—equivalent to merely 11.8% of the base model's original pre-training time (202 hours); for DiffBIR, the added cost is approximately 21% of the total training budget.

In exchange for this modest one-time investment, DiffusionReward delivers permanent and significant quality gains. For instance, OSEDiff (+Ours) reduces the FID score by 31.8% (from 65.13 to 44.40) and significantly improves identity consistency (LMD decreased by 0.48). These results demonstrate that our framework offers a highly favorable trade-off: achieving state-of-the-art perceptual quality and fidelity with scalable training costs and efficient, unchanged inference speeds.

Table 14: Comparison of Training Cost, Inference Speed, and Performance Metrics (Tested on NVIDIA L20 GPU). Values in parentheses indicate the absolute change compared to the base model. Arrows ($\downarrow$ / $\uparrow$) denote the direction of change. $\downarrow$ indicates lower is better (Improvement), $\uparrow$ indicates higher is better (Improvement).

| Method | Training Time | Inference Speed | Deg.$\downarrow$ | LMD$\downarrow$ | MUSIQ$\uparrow$ | FID$\downarrow$ |
|---|---|---|---|---|---|---|
| OSEDiff (Base) | 202 GPU hours | 0.13s | 46.20 | 2.8871 | 73.41 | 65.13 |
| OSEDiff (+Ours) | 24 GPU hours | 0.13s | 38.41 ($\downarrow$ 7.79) | 2.4060 ($\downarrow$ 0.48) | 75.24 ($\uparrow$ 1.83) | 44.40 ($\downarrow$ 20.73) |
| DiffBIR (Base) | 216 GPU hours | 2.84s | 35.16 | 2.2661 | 74.46 | 45.50 |
| DiffBIR (+Ours) | 46 GPU hours | 2.84s | 30.61 ($\downarrow$ 4.55) | 1.8642 ($\downarrow$ 0.40) | 74.82 ($\uparrow$ 0.36) | 42.59 ($\downarrow$ 2.91) |

## F    DISCUSSION ON REWARD HACKING IN BLIND FACE RESTORATION

Reward Hacking is a prevalent challenge in tasks employing Reward Feedback Learning (ReFL). Our research has found that Reward Hacking is also an issue in the BFR task. This phenomenon occurs when the generative model, in its pursuit of maximizing scores from a reward model, discovers and exploits "loopholes" or biases within the reward function. Such behavior, driven purely by score optimization, can lead to outputs that, despite achieving high reward scores, severely deviate from the desired effects of realistic, high-quality, and faithful restoration of the original input. This typically manifests as unnatural artifacts, stylistic distortions, or a loss of diversity. One of the core contributions of our work, particularly the dynamic updating strategy for the Face Reward Model (FRM), is specifically designed to mitigate such issues.

Fig. 12 (left) showcases examples of facial images generated during the face restoration task when Reward Hacking occurs. These examples reveal two distinct manifestations:

- **Style 1** represents a more severe form of Reward Hacking. In this scenario, the restored facial images exhibit a uniform, stylized, almost "painterly" appearance. Although certain features might appear sharp or well-defined, the overall output loses photorealism and may introduce exaggerated or unnatural facial characteristics. This suggests that the model has essentially learned a specific artistic style that the static reward model erroneously favors.
- **Style 2** reveals a significant yet different manifestation of Reward Hacking. In this case, the restored facial images consistently display unnatural blemishes, such as repetitive skin texture patterns, or exhibit a subtle "uncanny" appearance despite being overly smoothed. The emergence of these defects is likely because they inadvertently trigger higher scores from a less robust reward model, which may have failed to effectively penalize such subtle deviations from realism.

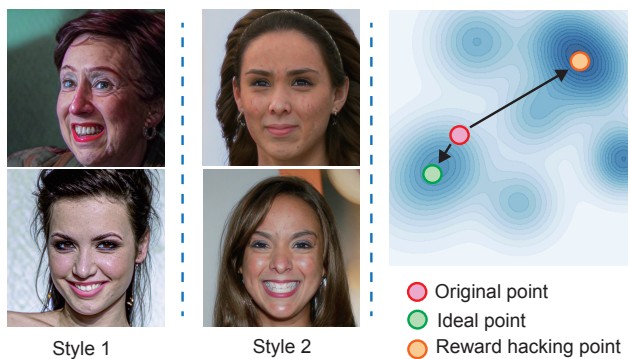

Figure 12: Illustration of Reward Hacking. (Left) Examples of facial restoration exhibiting reward hacking: Style 1 shows severe stylization, while Style 2 displays consistent artifacts and blemishes. (Right) A schematic representation in the image manifold space: The red point is the original output state. The orange point represents a reward hacking state, achieving high reward by moving off the natural image manifold. The green point indicates an ideal optimization outcome, improving reward while maintaining fidelity to the true manifold. Contour lines indicate reward values (darker is higher).

Fig. 12 (right) provides a schematic illustration of the Reward Hacking phenomenon within a conceptual image manifold space. The contour lines in the diagram represent the distribution of reward values, with darker blue areas indicating regions perceived by the reward model as having higher reward values.

- **Original point (red circle)** denotes the initial state of the model's output. This point is typically located on or near the true manifold of natural, realistic (facial) images, but its perceived quality may still be deficient.
- **Reward Hacking point (orange circle)** represents the outcome of an unconstrained or improperly guided optimization process. The model, by solely aiming to maximize the reward score, has moved to a high-reward region. However, this point is often distant from the initial state and, crucially, may have deviated from the manifold of realistic images. This occurs because the model exploits biases or vulnerabilities in the reward function, leading to outputs that, despite high scores, are perceptually flawed, overly stylized, or contain artifacts (as exemplified by Style 1 and Style 2).
- **Ideal point (green circle)**, in contrast, illustrates a more balanced and desirable optimization outcome. This state represents a moderate yet genuine improvement in reward/perceptual quality, while ensuring that the output remains close to the initial state and, most importantly, stays on or near the true manifold of natural, realistic images. This ensures the fidelity and realism of the results. Achieving this "green point" is the goal of robust ReFL frameworks, such as our proposed DiffusionReward method with its dynamic FRM updates, which actively counteracts overfitting to a static reward function and guides the restoration process towards genuine, manifold-consistent improvements.

Understanding and addressing Reward Hacking is crucial for developing reliable ReFL-based image restoration methods. Without effective countermeasures, the restoration model might merely learn to generate "reward-maximizing illusions" rather than truly enhancing the perceptual quality and faithfulness of the input images. Fortunately, by reducing the weight of the reward loss, using weight regularization, and employing an updatable face reward model, this issue can be alleviated or even resolved in practice; in our experiments, these strategies keep the optimization trajectory close to the natural face manifold and prevent the collapse behaviors illustrated in Fig. 12.

# G MORE ANALYSIS

## G.1 GENERALIZATION TO DIFFERENT MODEL ARCHITECTURES

To rigorously assess the generalizability of our proposed framework, we extended our evaluation to models with fundamentally different generative architectures. In addition to the diffusion-based base model, we integrated our method with two representative state-of-the-art approaches: the

Table 15: Generalization results on different architectures. We compare the integration of our method with GAN-based (GFPGAN), VQ-based (CodeFormer), and Diffusion-based (DR2) models. Best results for each model pairing are in **bold**.

| Model | LMD↓ | FID↓ | MUSIQ↑ |
|---|---|---|---|
| GFPGAN | 2.4110 | 42.57 | **73.90** |
| GFPGAN (+ours) | **2.4007** | **41.78** | 73.27 |
| CodeFormer | 1.9963 | 45.57 | **74.23** |
| CodeFormer (+ours) | **1.9943** | **38.77** | 70.12 |
| DR2 | 4.5449 | 62.54 | 70.19 |
| DR2 (+ours) | **3.2145** | **51.34** | **72.60** |

GAN-based GFPGAN (Chan et al., 2021) and the CodeFormer (Zhou et al., 2022), which utilizes a vector-quantized (VQ) codebook prior. We compared these against an alternative diffusion-based model, DR2 (Wang et al., 2023b). The performance of both the original and enhanced versions on the CelebA-Test dataset is detailed in Table 15.

The quantitative results reveal significant disparities in how different architectures respond to our Reward Feedback Learning (ReFL):

- Our method brings substantial improvements to the diffusion-based DR2 model across all key dimensions, including LMD, FID, and MUSIQ. This confirms that the stochastic generation process of diffusion models, characterized by the iterative injection of random noise, provides a broad and smooth exploration landscape. This inherent randomness is highly conducive to our framework, allowing the reward gradients to effectively guide the restoration trajectory toward the real face manifold.
- The optimization effect on GFPGAN is marginal, with most metrics showing negligible changes. We attribute this primarily to the deterministic nature of CNN-based GAN generators. The mapping from the latent code to the image is relatively rigid, resulting in a constrained "exploration space" that resists the fine-grained adjustments attempted by the reward feedback.
- For CodeFormer, ReFL improves fidelity and distributional alignment (LMD: 1.9963→1.9943; FID: 45.57→38.77) at the cost of perceptual quality (MUSIQ: 74.23→70.12). Like GFPGAN, its lack of intrinsic stochasticity prevents reward-guided exploration of diverse restorations.

Ultimately, this comparative experiment underscores that our framework exhibits the strongest synergy with stochastic generative models. While it can improve fidelity in deterministic or discrete architectures (like CodeFormer), it excels in the continuous and probabilistic solution space offered by diffusion models, where it can simultaneously enhance both fidelity and perceptual quality.

### G.2 DISCUSSION ON THE FACE REWARD MODEL'S ALIGNMENT WITH HUMAN PREFERENCES

Our Face Reward Model (FRM) is designed to capture subjective human preferences for face restoration, rather than simply predicting objective quality metrics. To this end, we employ a hybrid annotation strategy, leveraging a small amount of manually annotated preference data to build a domain-specific dataset for fine-tuning the general HPSv2 preference model. The optimized FRM achieves a consistency of **87.78%** with human judgments, significantly outperforming the baseline model's **63.05%**. This result strongly validates that the FRM is a true reward model aligned with subjective human perception, confirming its core role within the Reward Feedback Learning (ReFL) framework and ensuring that the optimization is guided by human aesthetic standards.

## H MORE QUALITATIVE RESULTS.

### H.1 QUALITATIVE RESULTS

Building upon the comparative results presented in Sec. 4.2 of the main paper, we provide further qualitative comparisons in this section. Figure 13 illustrates qualitative comparisons of our method

against other advanced methods on the synthetic CelebA-Test dataset. Similarly, Figure 14 showcases qualitative comparisons of our method with other advanced methods on real-world datasets.

## H.2 UNCURATED QUALITATIVE RESULTS

To showcase the effectiveness of our method in an unbiased manner, we consecutively selected the first 20 images from the CelebA-Test dataset for qualitative evaluation. The CelebA-Test dataset is derived from the public VQFR repository[4]. Specifically, we utilize a total of 20 images for restoration, ranging from `00000000.png` to `00000019.png`, located in the unzipped `celeba_512_validation_lq` directory. The comparison focuses on the baseline models, OSEDiff and DiffBIR, versus our enhanced variants, OSEDiff (+Ours) and DiffBIR (+Ours). The specific results are presented in Figure 15 and Figure 16.

# I  LLM USAGE STATEMENT

During the preparation of this paper, we utilized a LLM to assist with grammar correction and improving the clarity of our writing. We confirm that all scientific contributions, including ideation and analysis, are entirely the authors' original work. The LLM was used solely for proofreading purposes and did not contribute scientifically.

---

[4]https://github.com/TencentARC/VQFR

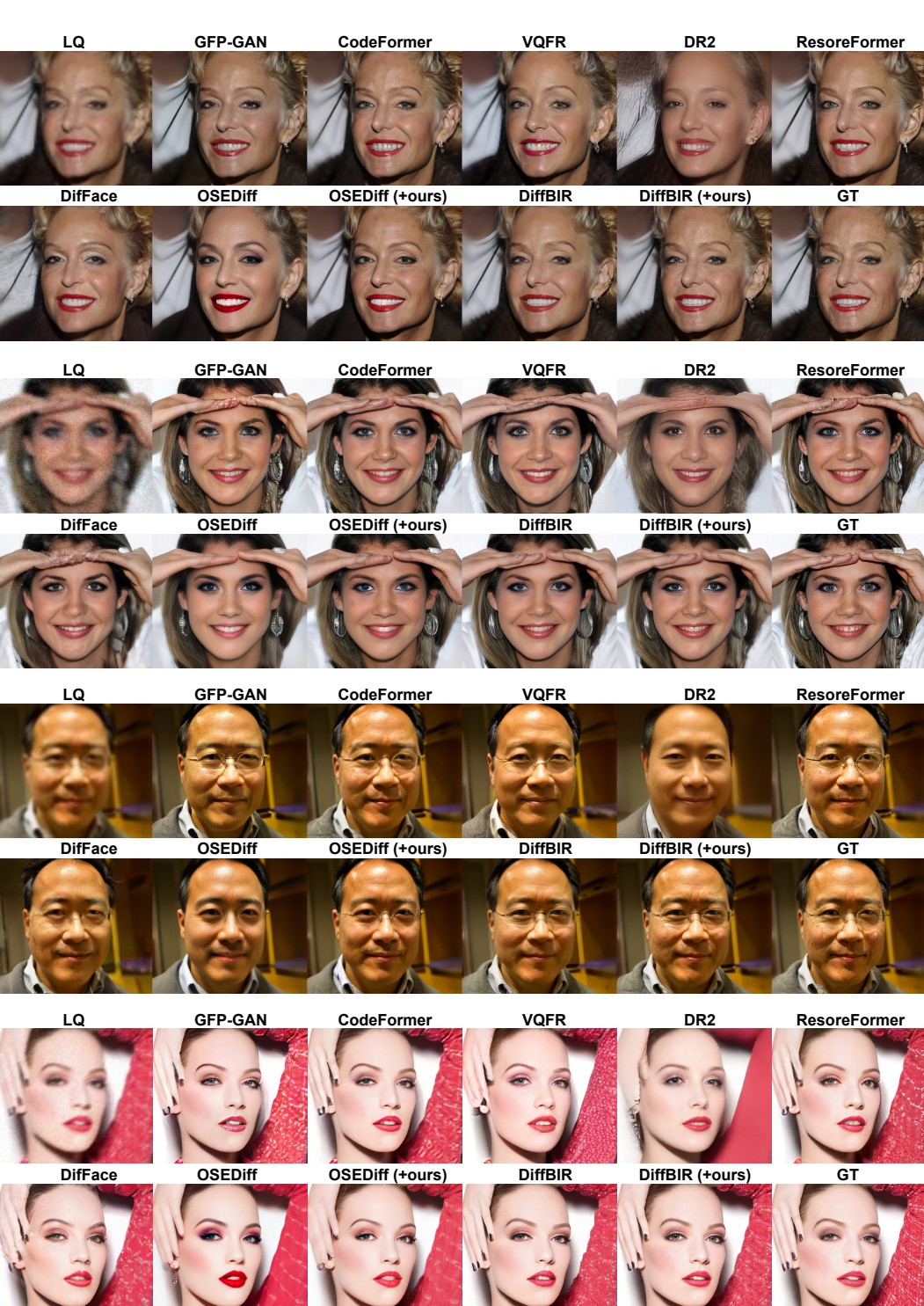

Figure 13: More qualitative comparison on the CelebA-Test. (Zoom in for details)

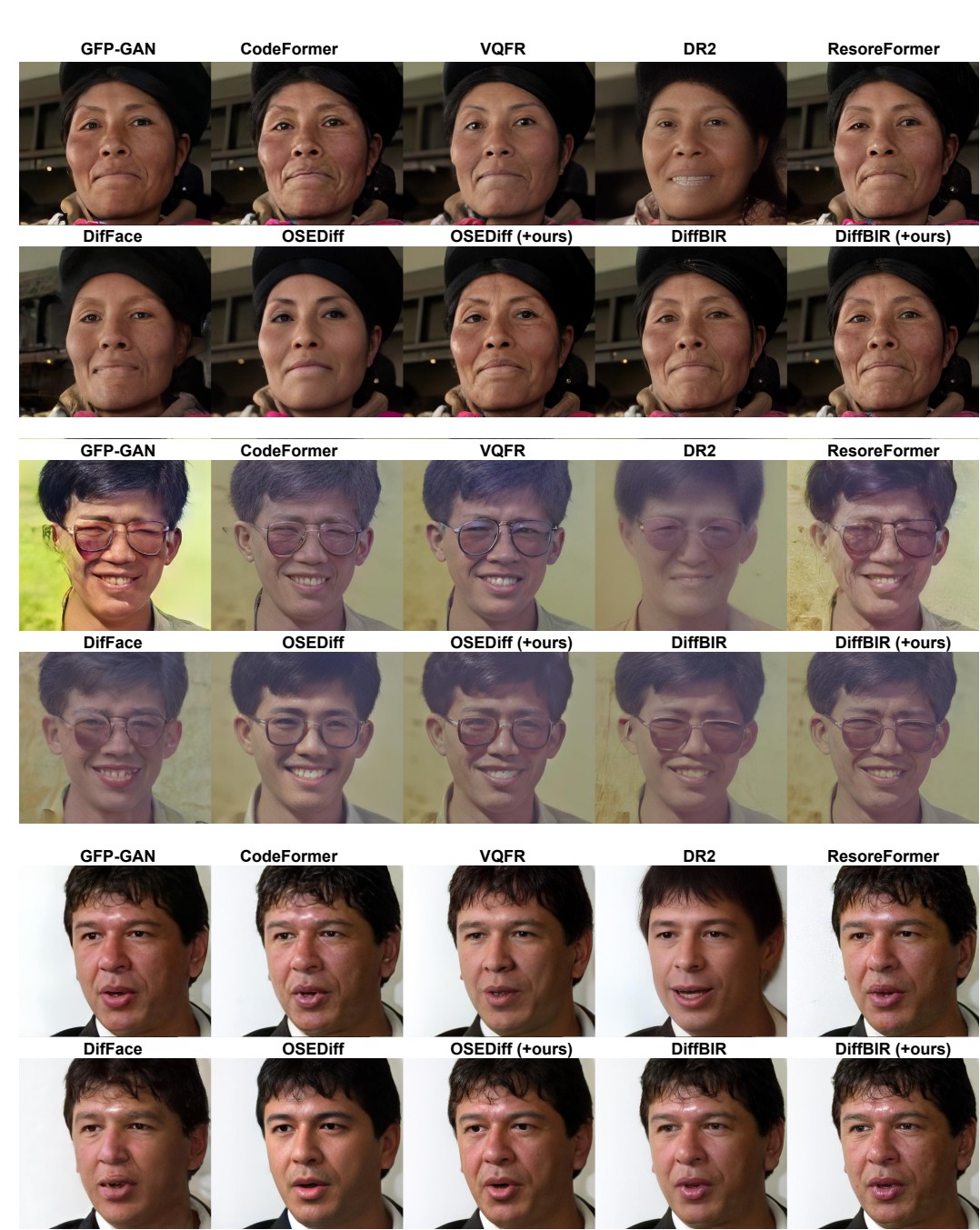

Figure 14: More qualitative comparison on the real-world faces. (Zoom in for details)

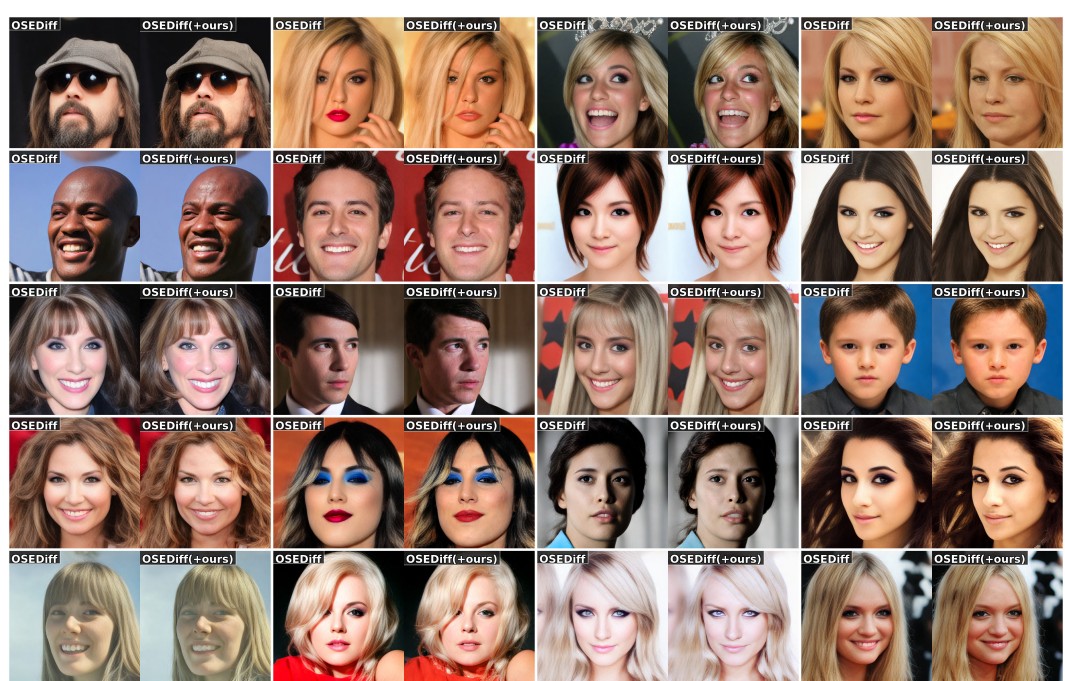

Figure 15: Qualitative comparison results between OSEDiff and OSEDiff (+Ours) without cherry-picking. (Zoom in for details)

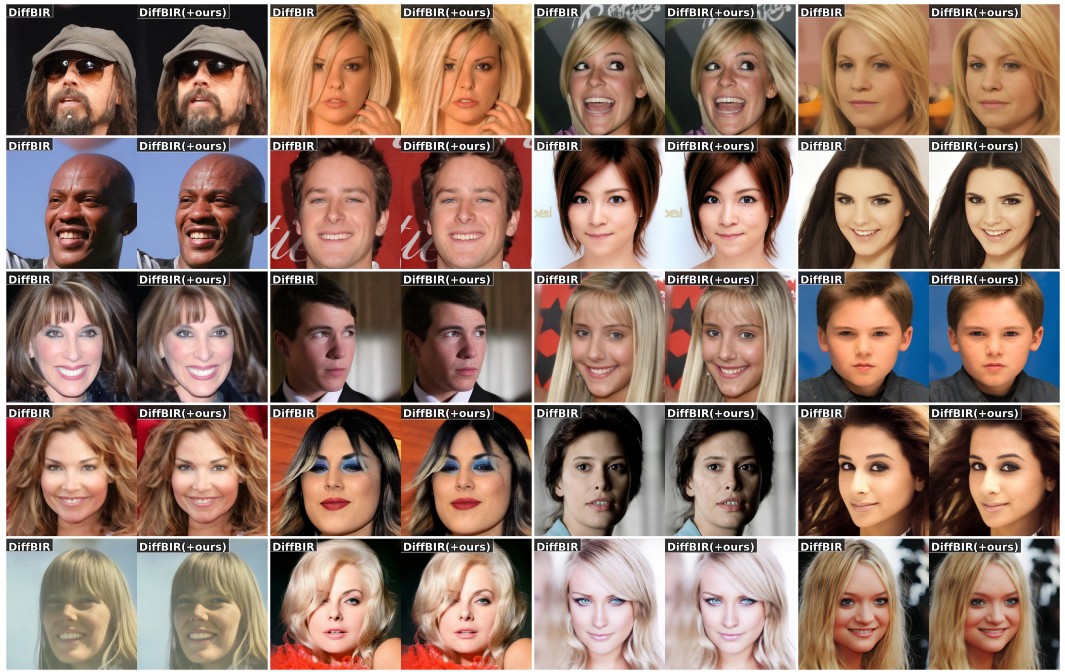

Figure 16: Qualitative comparison results between DiffBIR and DiffBIR (+Ours) without cherry-picking. (Zoom in for details)

