# OpenReview forum: "DiffusionReward: Enhancing Blind Face Restoration through Reward Feedback Learning"
_ICLR.cc/2026/Conference — Submitted to ICLR 2026_

### Official Review · Reviewer_HVjR · 2025-10-30

**Soundness:** 3
**Presentation:** 3
**Contribution:** 3
**Rating:** 4
**Confidence:** 4

**Summary:**

This paper introduces DiffusionReward, a framework applying Reward Feedback Learning (ReFL) to the Blind Face Restoration (BFR) task. The key idea is to train a Face Reward Model (FRM) that provides preference-based feedback to guide diffusion model optimization. The FRM is dynamically updated to avoid reward hacking. The framework is evaluated using DiffBIR and OSEDiff as base models and claims consistent improvements in both synthetic and real-world datasets in terms of identity preservation and perceptual quality.

**Strengths:**

1. The paper extends reinforcement-style reward optimization into the restoration domain, which is a cross-domain adaptation of alignment techniques.

2. Introducing online updating of the FRM to mitigate reward hacking is conceptually strong and practically meaningful.

3.The paper is easy to follow, with clear figures and a logical explanation of methodology.

**Weaknesses:**

1. Lack of analysis on reward accuracy and sparsity: Although the Face Reward Model (FRM) is central to the proposed framework, the paper lacks an in-depth analysis of reward reliability (accuracy with respect to human preference) and reward sparsity or noise sensitivity. The authors mention a hybrid human–SVM annotation pipeline, yet there is no quantitative study of how accurate or consistent the FRM’s feedback is compared to real human judgment, especially after dynamic updates. Moreover, ReFL optimization critically depends on reward signal density — sparse or unstable reward gradients can easily lead to optimization collapse or reward hacking.

2. Lack of convergence and training dynamics analysis，The authors claim that introducing the Face Reward Model (FRM) can provide effective gradient feedback to guide optimization. Intuitively, this should accelerate convergence or stabilize training. However, the paper does not present any training dynamics analysis, such as loss curves, reward evolution, or convergence speed comparisons with baseline models.

3. Diffusion models are known to be slow. The paper does not report inference speed or resource cost, nor analyze trade-offs introduced by ReFL fine-tuning.

4. The proposed ReFL framework introduces supervision at the image (decoder) level by backpropagating through the image reward model. This incurs significant training-time and memory overhead, as gradients must flow through the decoder and multiple denoising steps. Although the authors later mention truncated backpropagation (N=1), there is no quantitative analysis of the actual training cost compared with the base models. This raises concerns about scalability and practicality.

5. The related work section overlooks several important prior approaches:

[1].Face Super-Resolution Guided by 3D Facial Priors

[2]. Rethinking Deep Face Restoration

**Questions:**

CodeFormer and related transformer-based BFR methods already demonstrate strong perceptual quality and robustness. The current ReFL framework is tied to the diffusion process, where gradient feedback is injected during denoising. It remains unclear whether the proposed reward learning paradigm is architecture-agnostic or only applicable to diffusion-based models.

---

> ### Author Response · Authors · 2025-11-23
> **Response to Reviewer HVjR Part 1/4**
>
> We thank you for your expert feedback and for recognizing the conceptual and practical significance of our work. Regarding your concerns, we have revised and uploaded the manuscript, with all changes highlighted in blue for easy reference. Next, we will reply point by point.
>
> ### W1:
>
> > *“Lack of analysis on reward accuracy and sparsity: Although the Face Reward Model (FRM) is central to the proposed framework, the paper lacks an in-depth analysis of reward reliability (accuracy with respect to human preference) and reward sparsity or noise sensitivity. The authors mention a hybrid human–SVM annotation pipeline, yet there is no quantitative study of how accurate or consistent the FRM’s feedback is compared to real human judgment, especially after dynamic updates. Moreover, ReFL optimization critically depends on reward signal density — sparse or unstable reward gradients can easily lead to optimization collapse or reward hacking.”*
>
> **Response:**
>
> We appreciate the reviewer's insightful analysis and agree that reward reliability and stability are essential for the success of  ReFL.  We then respectfully offer three points of clarification.
>
>
> Firstly, we would like to clarify the point raised by the reviewer:
>
> > "There is no quantitative study of how accurate or consistent the FRM’s feedback is compared to real human judgment."
>
> As mentioned in **Section 3.2 (first paragraph)** of our original submission, we included a quantitative analysis of reward accuracy, specifically its alignment with human judgment. Our Face Reward Model (FRM) achieves **87.78% human consistency**, surpassing the baseline model HPSv2, which achieves **63.05%**.
>
> Additionally, we conducted human consistency experiments comparing models with and without textual input. The model with text-based rewards outperforms the non-textual version, with human consistency ratings of **87.78% vs. 85.01%**, respectively. These results are further detailed in **Appendix D.4**.
>
> Furthermore, the user study in **Table 6** demonstrates that human evaluators overwhelmingly prefer our restored images over the baselines in terms of both fidelity and realism.
>
> **Table 6: Human preference evaluation**
>
> | Comparison                     | Fidelity Preference | Realism Preference |
> | ------------------------------ | ------------------- | ------------------ |
> | OSEDiff (+ours) **vs** OSEDiff | **78%** vs 22%      | **88%** vs 12%     |
> | DiffBIR (+ours) **vs** DiffBIR | **68%** vs 32%      | **75%** vs 25%     |
>
>
> Next, we would like to clarify the point raised by the reviewer:
>
> > especially after dynamic updates
>
> Regarding the reviewer's specific concern about reward reliability "especially after dynamic updates," we additionally measure **Human Consistency** of the FRM at different update iterations on a held-out test set of 360 annotated pairs. The results are presented in the newly added **Table 10**:
>
> | Iteration                 | 0          | 500    | 1000   | 1500   | 2000   | 2500   | 3000   |
> | :------------------------ | :--------- | :----- | :----- | :----- | :----- | :----- | :----- |
> | **Human Consistency (↑)** | **87.78%** | 86.11% | 85.83% | 85.00% | 83.89% | 83.01% | 83.06% |
>
> As shown in **Table 10**, human consistency remains high (>83%) and stable, **significantly surpassing the HPS v2 baseline (~63%)**. The slight decline is a **deliberate trade-off** designed to prevent **Reward Hacking**.  By dynamically anchoring the reward model to the **real face manifold** (Section 3.3, Eq. 8), we strictly penalize unrealistic "high-score" artifacts (as analyzed in **Appendix F**). This mechanism sacrifices a marginal amount of generic consistency to "tighten" the reward boundary, ensuring the feedback signal effectively rejects hacked results and guides the model toward true photorealism.
>
> Finally, we would like to clarify the point raised by the reviewer:
>
> > ReFL optimization critically depends on reward signal density — sparse or unstable reward gradients can easily lead to optimization collapse or reward hacking.
>
> We fully agree with the reviewer regarding this critical issue. In fact, we have analyzed the problem of 'sparse or unstable reward signals leading to reward hacking' across multiple sections of our original manuscript.
>
> We have observed this phenomenon in the BFR task and provided a detailed analysis of its causes and mechanisms, along with visual examples of hacking, in **Appendix F**.
>
> To mitigate the instability caused by reward noise, we proposed the **Dynamic Reward Update** mechanism in **Section 3.3**. By dynamically anchoring the optimization to the **real face manifold**, this strategy effectively prevents the model from collapsing into "hacked" solutions. This is validated in **Section 4.3**, where **Variant 2** (without dynamic updates) clearly exhibits reward hacking artifacts. Furthermore, as noted in **Appendix F**, We also emphasize that weight regularization and smaller reward loss weight helps to mitigate this issue.

---

> > ### Author Response · Authors · 2025-11-23
> > **Response to Reviewer HVjR Part 2/4**
> >
> > ### W2. Convergence and training dynamics analysis
> >
> > > *“Lack of convergence and training dynamics analysis: The authors claim that introducing the Face Reward Model (FRM) can provide effective gradient feedback to guide optimization. Intuitively, this should accelerate convergence or stabilize training. However, the paper does not present any training dynamics analysis, such as loss curves, reward evolution, or convergence speed comparisons with baseline models.”*
> >
> > **Response:**
> >
> > We agree that it is important to verify that the ReFL training is well behaved. We clarify below our claims and the additional analyses included in the revised version.
> >
> > 1. **On “accelerating convergence” vs. our actual goal.**
> >    Our use of the Face Reward Model (FRM) is intended to provide an additional, preference-aligned gradient signal to *refine* pre-trained diffusion-based restoration models, rather than to accelerate or fundamentally change their convergence behavior. As discussed in the Introduction and **Sec. 3**, we do not train diffusion models from scratch; instead, we perform ReFL fine-tuning on strong BFR baselines (DiffBIR and OSEDiff) to address specific limitations in high-frequency facial details and identity consistency. Therefore, our claims focus on **improving the final restoration quality** rather than on faster convergence compared to the original pre-training.
> >
> > 2. **Training dynamics analysis (added in the revision).**
> >    In response to the reviewer’s suggestion, we have added an training-dynamics analysis in the revised manuscript. In **Fig. 7**, we now report both the evolution of the structural consistency loss and the FRM reward over ReFL fine-tuning iterations.

---

> ### Author Response · Authors · 2025-11-23
> **Response to Reviewer HVjR Part 3/4**
>
> ### W3 & W4. Training and inference overhead
>
> > *“Diffusion models are known to be slow. The paper does not report inference speed or resource cost, nor analyze trade-offs introduced by ReFL fine-tuning.”*
>
> > *“The proposed ReFL framework introduces supervision at the image (decoder) level by backpropagating through the image reward model. This incurs significant training-time and memory overhead, as gradients must flow through the decoder and multiple denoising steps. Although the authors later mention truncated backpropagation (N=1), there is no quantitative analysis of the actual training cost compared with the base models. This raises concerns about scalability and practicality.”*
>
> **Response:**
> You misunderstood our approach. We would like to clarify that ReFL is a post-training refinement approach , whose training and inference overhead is typically dependent on the complexity of the pre-trained base restoration model.
> Following your suggestion, we have additionally provided a quantitative analysis of both inference latency and training-time overhead for our ReFL fine-tuning, with the results summarized in **Table 14**.
>
> 1. **Inference speed and resource cost.**
>    Our ReFL framework is applied **only during the fine-tuning phase**. At test time, the Face Reward Model (FRM) and all gradient computations are **not used**, so the inference pipeline is exactly the same as the original BFR models. Consequently, **inference speed and resource usage remain identical to the base models**. For example, OSEDiff (+Ours) and DiffBIR (+Ours) have the same inference times as their respective base models (0.13s and 2.84s per image on an NVIDIA L20), as shown in Table R1. Our method introduces **no additional latency** during inference.
>
> 2. **Training cost and trade-offs.**
>    ReFL is designed as a **cost-effective post-training refinement**. We fine-tune from pre-trained weights rather than training diffusion models from scratch, and we employ truncated backpropagation with $N=1$ to control the training overhead. In this setting, gradients only pass through the last denoising step (the decoder is frozen), instead of all $T$ denoising steps, which keeps both memory usage and per-iteration time moderate.
>
>    As reported in Table 14, fine-tuning OSEDiff (+Ours) requires only **24 GPU hours**, which is about **11.8%** of the base model’s original training time (202 hours). For DiffBIR (+Ours), the additional fine-tuning cost is only **21%** of the original training time. Despite this relatively small one-time cost, our method yields substantial and permanent improvements in restoration quality. For example, on OSEDiff, FID is reduced from **65.13 → 44.40**, and identity consistency is significantly improved (LMD from **2.8871 → 2.4060**).
>
>    Overall, this shows that DiffusionReward is practical to train in terms of both time and memory, and that it offers a favorable cost–benefit ratio.
>
> **Table 14: Comparison of training cost, inference speed, and performance metrics (NVIDIA L20 GPU 48 GB)**
> **↓ indicates lower is better, ↑ indicates higher is better.**
>
> | Method          | Training Time | Inference Speed | Deg. ↓         | LMD ↓           | MUSIQ ↑        | FID ↓           |
> | --------------- | ------------- | --------------- | -------------- | --------------- | -------------- | --------------- |
> | OSEDiff (Base)  | 202 GPU hours | 0.13s           | 46.20          | 2.8871          | 73.41          | 65.13           |
> | OSEDiff (+Ours) | 24 GPU hours  | 0.13s           | 38.41 (↓ 7.79) | 2.4060 (↓ 0.48) | 75.24 (↑ 1.83) | 44.40 (↓ 20.73) |
> | DiffBIR (Base)  | 216 GPU hours | 2.84s           | 35.16          | 2.2661          | 74.46          | 45.50           |
> | DiffBIR (+Ours) | 46 GPU hours  | 2.84s           | 30.61 (↓ 4.55) | 1.8642 (↓ 0.40) | 74.82 (↑ 0.36) | 42.59 (↓ 2.91)  |
>
> The above discussion has been included in **Appendix E** and is highlighted in blue font.

---

> ### Author Response · Authors · 2025-11-23
> **Response to Reviewer HVjR Part 4/4**
>
> ### W5. Missing related work in face restoration
>
> > *“The related work section overlooks several important prior approaches:
> >
> > [1]. Face Super-Resolution Guided by 3D Facial Priors
> > [2]. Rethinking Deep Face Restoration”*
>
> **Response:**
>
> Thank you for your recommendation. These are two excellent works. I have carefully read them and included them in our related work section. That seriously strengthened the background.
>
> ------
>
> ### Q1. Architecture-agnostic applicability beyond diffusion models
>
> > *“CodeFormer and related transformer-based BFR methods already demonstrate strong perceptual quality and robustness. The current ReFL framework is tied to the diffusion process, where gradient feedback is injected during denoising. It remains unclear whether the proposed reward learning paradigm is architecture-agnostic or only applicable to diffusion-based models.”*
>
> **Response:**
> In fact, ReFL itself was originally designed for diffusion models, and its effectiveness is significantly more pronounced on diffusion-based methods. The reason is that diffusion models are a type of **probabilistic generative model**, and their denoising process inherently involves stochasticity, which facilitates exploration by the reward model. This characteristic has also been widely recognized in T2I tasks as well as RL for LLMs. The starting point of our paper is to address the limitations of existing diffusion-based restoration methods, as can be seen from the original text in the Introduction:
>
> "However, these pre-trained diffusion models typically undergo training using images from general domains, which lack an adequate amount of face-specific prior knowledge. This deficiency frequently gives rise to restored facial images that are short of detailed features."
>
> Moreover, in our paper, we model the process based on the denoising procedure of diffusion models and perform optimization via back-propagating gradients; the theoretical foundation is also rooted in diffusion models. Our experiments further validate that post-training DiffBIR and OSEDiff with DiffusionReward leads to substantial performance gains.
>
> In response to the reviewer’s request, we attempted to forcibly adapt our framework to CodeFormer and additionally tested it on DR2 (diffusion-based) and GFPGAN (convolutional architecture):
>
> | Model                  | LMD ↓   | FID ↓   | MUSIQ ↑ |
> |------------------------|---------|---------|---------|
> | GFPGAN                 | 2.4110  | 42.57   | 73.90   |
> | GFPGAN (+ours)         | 2.4007  | 41.78   | 73.27   |
> | CodeFormer             | 1.9963  | 45.57   | 74.23   |
> | CodeFormer (+ours)     | 1.9943  | 38.77   | 70.12   |
> | DR2                    | 4.5449  | 62.54   | 70.19   |
> | DR2 (+ours)            | 3.2145  | 51.34   | 72.60   |
>
> We observe that:
>
> * On the diffusion model DR2, our method brings substantial improvements in all key metrics (LMD, FID, MUSIQ), demonstrating the strong synergy between our framework and diffusion models.
>
> * On CodeFormer, our method improves LMD and FID.
>
> * On GFPGAN, the effect is more modest, with slight improvements in all metrics.
>
> The results show that DiffusionReward achieves significant improvements on diffusion-based restoration methods (DR2), and when forcibly transplanted to CodeFormer, it still yields improvements in two metrics—especially a notable gain in FID. This is consistent with the claims in our paper: our approach delivers substantial enhancements for diffusion-based restoration methods while providing certain gains for restoration models with other architectures.

---

### Official Review · Reviewer_nDxx · 2025-10-30

**Soundness:** 3
**Presentation:** 2
**Contribution:** 2
**Rating:** 6
**Confidence:** 3

**Summary:**

This paper identifies that pre-trained diffusion models for blind face restoration (BFR) often lack face-specific priors, leading to results deficient in detail and identity preservation. The authors propose a novel solution by adapting Reward Feedback Learning (ReFL) to BFR. Their key innovation involves fine-tuning a restoration model using a tailored, dynamically updated face reward model, combined with structural consistency and weight regularization constraints. This approach enhances the visual fidelity of restored faces while ensuring identity consistency.

**Strengths:**

1. Effective Adaptation:​​ The proposed application of Reward Feedback Learning (ReFL) to Blind Face Restoration (BFR) is novel and compelling. It presents a computationally efficient fine-tuning strategy that demonstrably enhances the performance of powerful pre-trained diffusion models, as evidenced by the clear improvements in both visual quality and quantitative metrics reported.

2. Well-Designed Reward Model:​​ The construction of the Face Reward Model (FRM) is a significant contribution. The meticulous data collection and annotation process for training the FRM is well-motivated and interesting.

3. ​Good Reproducibility:​​ The release of the source code, coupled with the extensive details provided in the appendix, facilitates future research and practical application, adhering to good practice in the research community.

4. ​Clarity and Readability:​​ The manuscript is clearly written and is easy to follow.

**Weaknesses:**

1. Potential Cherry-Picking of Qualitative Results:​​ A common concern in image restoration is the potential for cherry-picking qualitative results. While the provided examples are compelling, it would significantly strengthen the validity of the claims if the supplementary material included a more extensive set of randomly selected samples (e.g., the first 1-20 images from a standard benchmark like FFHQ) comparing the base models (OSEDiff, DiffBIR) against their fine-tuned versions with the proposed method. This would provide a more objective and convincing demonstration of the method's consistent performance gains.

2. ​Clarification of Novelty Relative to ReFL in Image Generation:​​ The manuscript would benefit from a more precise discussion of its novelty concerning the application of ReFL. Given the existing body of work applying ReFL to text-to-image and other generative tasks, the claim that "there remains a notable research gap in exploring the application of ReFL to restoration tasks" (Line 171) requires stronger justification. The authors should more clearly articulate the distinct challenges of adapting ReFL specifically for restoration(e.g., the critical importance of identity preservation and fidelity to the degraded input, as opposed to open-ended generation) and how their work addresses these unique challenges, thereby delineating their contribution beyond a direct transfer of the ReFL paradigm.

3. ​Correction of Presentation Errors:​​ There are minor presentation issues that should be corrected. For instance, in Figure 5, the image for "OSEDiff (+ours)" in the right set appears to be identical to the one labeled for "DiffBIR." A careful proofreading of all figures and captions is recommended to ensure accuracy.

**Questions:**

Please see Weaknesses.

---

> ### Author Response · Authors · 2025-11-23
> **Response to Reviewer nDxx**
>
> We thank you for recognizing the novelty of our work and its significant contribution to the community. Regarding your concerns, we have revised and uploaded the manuscript, with all changes highlighted in blue for easy reference. We will respond to your concerns point by point below.
>
> ### W1. Potential cherry-picking in qualitative comparisons
>
> > *“Potential Cherry-Picking of Qualitative Results: A common concern in image restoration is the potential for cherry-picking qualitative results. While the provided examples are compelling, it would significantly strengthen the validity of the claims if the supplementary material included a more extensive set of randomly selected samples (e.g., the first 1–20 images from a standard benchmark like FFHQ) comparing the base models (OSEDiff, DiffBIR) against their fine-tuned versions with the proposed method.”*
>
> **Response:**
>
> Your suggestion enables us to better showcase the superior performance of our method. Since FFHQ is part of our training data, showing the first images from FFHQ would mainly reflect performance on the training distribution. Instead, we follow the spirit of the suggestion by adding to the **Appendix H.2** a non-cherry-picked set of test images.: we consecutively select the first 20 images (*00000000.png*–*00000019.png*) from the CelebA-Test split in the public VQFR repository and compare OSEDiff / DiffBIR with their fine-tuned versions OSEDiff (+Ours) and DiffBIR (+Ours). The corresponding qualitative results are provided in **Figures 15 and 16**, demonstrating consistent improvements without cherry-picking.
>
> ------
>
> ### W2. Clarifying novelty relative to ReFL in image generation
>
> > *“Clarification of Novelty Relative to ReFL in Image Generation: The manuscript would benefit from a more precise discussion of its novelty concerning the application of ReFL. Given the existing body of work applying ReFL to text-to-image and other generative tasks, the claim that ‘there remains a notable research gap in exploring the application of ReFL to restoration tasks’ (Line 171) requires stronger justification. The authors should more clearly articulate the distinct challenges of adapting ReFL specifically for restoration (e.g., the critical importance of identity preservation and fidelity to the degraded input, as opposed to open-ended generation) and how their work addresses these unique challenges, thereby delineating their contribution beyond a direct transfer of the ReFL paradigm.”*
>
> **Response:**
>
> We have incorporated this discussion into the related work. Please refer to the **Sec. 2** for the details. We thank the reviewer for this suggestion, which has substantially enriched our related work discussion.
>
> You can find the specific added content on page 4 (highlighted in blue):
>
> *While existing ReFL paradigms succeed in open-ended text-to-image synthesis, their direct application to image restoration is constrained by the precise face assessment and strict identity maintenance. We overcome these limitations by incorporating two key refinements to the ReFL framework: (i) a specialized Face Reward Model (FRM) for accurate facial quality assessment, and (ii) an structural consistency constraint to enforce identity preservation. Furthermore, we implement an innovative dynamic updating mechanism to effectively mitigate reward hacking, thereby yielding a substantial elevation in overall restoration quality.*
>
> ------
>
> ### W3. Presentation errors and figure corrections
>
> > *“Correction of Presentation Errors: There are minor presentation issues that should be corrected. For instance, in Figure 5, the image for ‘OSEDiff (+ours)’ in the right set appears to be identical to the one labeled for ‘DiffBIR.’ A careful proofreading of all figures and captions is recommended to ensure accuracy.”*
>
> **Response:**
>
> We have carefully proofread the content and corrected this issue. The revised version is reflected in **Figure 5.**

---

### Official Review · Reviewer_4Q9v · 2025-10-31

**Soundness:** 2
**Presentation:** 3
**Contribution:** 2
**Rating:** 4
**Confidence:** 4

**Summary:**

The paper proposes DiffusionReward, a Reward Feedback Learning (ReFL) framework for blind face restoration (BFR). The key idea is to introduce a Face Reward Model (FRM)—a CLIP-based reward network fine-tuned with a hybrid of human and automatically annotated face-preference data—to provide perceptual feedback during the optimization of diffusion-based restoration models (e.g., DiffBIR, OSEDiff). Experiments on synthetic and real-world datasets demonstrate consistent improvements over state-of-the-art BFR methods on multiple perceptual and aesthetic metrics.

**Strengths:**

1. A dynamic reward update strategy to mitigate reward hacking;

2. Structural consistency and weight-regularization constraints to preserve identity and maintain generative diversity;

3. End-to-end ReFL training integrated into the denoising process.

**Weaknesses:**

1. The core contribution of this work is to introduce the reward feedback mechanism into BFR. However, the experimental results do not clearly demonstrate the necessity and advantages of the reward model. Specifically focusing on the ablation studies of Table 4, i) the LMD improvement mainly comes from the structural consistency loss (see from Base and Variant 1); ii) the MUSIQ and Aesthetic improvement is primarily controlled by KL weight regularization (see from Variant 3 and Ours) as this KL regularization encourages sufficiently using the Diffusion prior; iii) I suggest the author supply another experiment by changing the WR loss importance in Variant 1, i.e., adjusting the hyper-parameter $\lambda_{reg}$ to validate the function of the reward model.

2. This work attempts to optimize the entire multi-step denoising process (i.e., $N>1$) as claimed in Appendix D.2. However, the chain length $N$ is finally truncated to 1, thus being the same as the commonly used independent denoising strategy in Diffusion models. So, I can't capture the significance of the related discussion (e.g., Appendix D.2) in this work.

3. The reward classifier is pre-trained on some human-annotated data. How many annotated pairs are used? It would be better to analyze the sensitivity of the reward model and restoration model regarding the human annotation.

4. The reward model introduces the additional text prompt. I wonder about the significance of this textual information.

**Questions:**

See weakness.

---

> ### Author Response · Authors · 2025-11-23
> **Response to Reviewer 4Q9v Part 1/3**
>
> We sincerely thank the reviewer for the detailed analysis and suggestions. Regarding your concerns, we have revised and uploaded the manuscript, with all changes highlighted in blue for easy reference. We respond to each weakness and question point-by-point below.
>
> ### W1. Necessity and contribution of the reward model vs. loss terms in Table 4
>
> > *“The core contribution of this work is to introduce the reward feedback mechanism into BFR. However, the experimental results do not clearly demonstrate the necessity and advantages of the reward model. Specifically focusing on the ablation studies of Table 4, i) the LMD improvement mainly comes from the structural consistency loss (see from Base and Variant 1); ii) the MUSIQ and Aesthetic improvement is primarily controlled by KL weight regularization (see from Variant 3 and Ours) as this KL regularization encourages sufficiently using the Diffusion prior; iii) I suggest the author supply another experiment by changing the WR loss importance in Variant 1, i.e., adjusting the hyper-parameter λ to validate the function of the reward model.”*
>
> **Response:**
>
> We thank the reviewer for this insightful comment. We agree that Weight Regularization (WR) is crucial for preserving the diffusion prior, but we respectfully clarify that the improvements in MUSIQ and Aesthetic are **not** solely due to WR. Rather, they arise from the **synergistic coupling** between the Reward Model (which drives perceptual quality) and WR (which stabilizes the optimization). Below we provide clarifications and additional experiments.
>
> **1. Roles of WR vs. Reward Model**
>
> **WR as the anchor; reward as the driver.**
> By comparing Variant 1 (SC + WR only) and Variant 2 (SC + WR + Reward) in Table 4, we can directly see the contribution of the Reward Model. Variant 1 achieves a low MUSIQ of **54.70**, indicating that WR alone (even with structural consistency) tends to produce over-smoothed results with limited perceptual quality. Simply adding the Reward Model (Variant 2) raises MUSIQ to **71.12**, a substantial improvement that cannot be explained by WR alone. This suggests that WR mainly stabilizes the optimization, while the Reward Model is the key factor for enhancing perceptual quality.
>
> **Why Variant 3 drops.**
> The performance degradation of Variant 3 (w/o WR) compared to Ours does not mean that WR “creates” high quality. Instead, it shows that without WR, the optimization drifts too far from the pre-trained diffusion prior (catastrophic forgetting), weakening the model’s generative ability. The strong performance of “Ours” is therefore due to the **combination**: WR keeps the model within a valid diffusion manifold, and the Reward Model then effectively refines details and aesthetics.
>
> **2. New sensitivity analysis on WR (as requested)**
>
> To further validate that WR alone cannot achieve the reported performance, we followed the reviewer’s suggestion and conducted a sensitivity analysis on the hyper-parameter $\lambda_{\text{reg}}$ in Variant 1 (i.e., without the Reward Model). We tested $\lambda_{\text{reg}} \in \{10^{-3}, 10^{-4}\text{ (default)}, 10^{-5}\}$.
>
> **Table: Sensitivity of $\lambda_{\text{reg}}$ on Variant 1 (w/o Reward Model)**
>
> | **$\lambda_{\text{reg}}$** | **LMD (↓)** | **MUSIQ (↑)** | **Aesthetics (↑)** |
> | -------------------------- | ----------- | ------------- | ------------------ |
> | $10^{-3}$                  | 2.0252      | 57.57         | 5.7358             |
> | $10^{-4}$ (default)        | 1.9583      | 54.70         | 5.6572             |
> | $10^{-5}$                  | 1.9087      | 55.05         | 5.7729             |
>
> As shown, tuning WR in the absence of the Reward Model brings only minor changes and keeps MUSIQ in the range **54–57**, far below the **74.82** achieved by our full method. This confirms that WR by itself cannot recover the perceptual and aesthetic gains, and that the Reward Model is essential for high-quality restoration, while WR primarily acts as a stabilizer.
>
> These results and  discusion have been included in **Appendix D.6** to give readers a clearer picture of the respective roles of KL weight regularization and the Reward Model.

---

> ### Author Response · Authors · 2025-11-23
> **Response to Reviewer 4Q9v Part 2/3**
>
> ### W2. Is our N=1 setting equivalent to standard independent per-timestep training
>
> > *“This work attempts to optimize the entire multi-step denoising process (i.e., 𝒯) as claimed in Appendix D.2. However, the chain length N is finally truncated to 1, thus being the same as the commonly used independent denoising strategy in Diffusion models. So, I can't capture the significance of the related discussion (e.g., Appendix D.2) in this work.”*
>
> **Response:**
>
> We thank the reviewer for raising this important point. You misunderstood our approach. We wish to clarify that **$N$ represents the gradient backpropagation horizon, not the inference chain length.**
>
>  This setting (i.e. $N=1$) is fundamentally different from the standard independent denoising strategy. Below, we explain our training mechanism and its distinction from standard methods:
>
> **1. Our Training Mechanism (Full denoising Chain + Truncated Gradient):**
> In our ReFL fine-tuning stage, each training iteration executes the **full denoising chain** (all $T$ steps, e.g., $z_T \to \dots \to z_0 \to x_0$) in the forward pass to generate the final restored image. The optimization objective is a **reward-driven loss** defined directly on this final image. To update the model, we calculate the gradient of this final reward loss and backpropagate it. "Truncation $N=1$" simply means we stop the gradient flow after passing through the last denoising step (analogous to Truncated BPTT), rather than propagating it all the way back to $z_T$ to save memory.
>
> **2. Crucial Difference from Standard Independent Denoising:**
> Even with $N=1$, our approach is **not** equivalent to the standard per-timestep training used in Diffusion Models (e.g., DDPM).
> * **Input Distribution (Idealized vs. Actual State):**
>     * **Standard Training:** The input latent $z_t$ is sampled directly from the **Ground Truth image** by adding noise (i.e., $z_t = \sqrt{\bar{\alpha}_t}x_0 + \sqrt{1-\bar{\alpha}_t}\epsilon$). This represents an "idealized" noisy state. The model is trained independently at each step and never sees its own previous errors.
>     * **Our ReFL Training:** The input latent $z_{t_1}$ is the **actual output** from the model's previous denoising step (i.e., $z_{t_2} \xrightarrow{g_\theta} z_{t_1}$). This input implicitly contains the artifacts and accumulated errors produced by the model during the specific inference chain. Consequently, the model is optimized to correct its own generated "imperfect" latent to maximize the quality of the final image $x_0$.
> * **Loss Objective:** Standard training uses a local noise-matching loss (MSE). Our method uses a perceptual reward loss on the final restored image $x_0$, which requires the full reverse chain to be executed to be computed.
>
> **3. Significance of Appendix D.2 (Now Sec. 4.3):**
> The discussion in Appendix D.2 is significant because it provides the **conceptual formulation** of the restoration process as a unified, differentiable generator. It empirically justifies our design choice: we demonstrate that while full backpropagation is theoretically possible, expanding the gradient horizon $N$ beyond 1 yields diminishing returns in performance while drastically increasing computational cost. This matches prior observations in ReFL for diffusion models [1] (e.g., Clark et al., 2024). Thus, $N=1$ is a deliberate trade-off verified by the ablation study, not a fallback to standard independent training.
>
> We have modified the expression in three places to avoid potential ambiguity.:
> (i) in **Sec. 3.2**, we now state that **Eq. (1)** is only the standard latent diffusion objective used to pretrain the base BFR models, and it is not used during our ReFL fine-tuning stage;
> (ii) in **Sec. 3.3**, we explicitly describe our training procedure as running the full reverse denoising chain in the forward pass while applying truncated backpropagation with $N = 1$ on the reward-driven loss (plus structural and regularization terms) defined on the final restored image; and
> (iii) in **Sec. 4.3 and Table 5 (Original Appendix D.2)**，We have simplified the discussion by removing the description of full BPTT, now focusing solely on the ablation study of gradient truncation length.
>
> [1] Clark, K., Vicol, P., Swersky, K., & Fleet, D. J. (2024). Directly Fine-Tuning Diffusion Models on Differentiable Rewards. *ICLR 2024*.

---

> > ### Author Response · Authors · 2025-11-23
> > **Response to Reviewer 4Q9v Part 3/3**
> >
> > ### W3. Human-annotated data scale & sensitivity to annotation
> >
> > > *“The reward classifier is pre-trained on some human-annotated data. How many annotated pairs are used? It would be better to analyze the sensitivity of the reward model and restoration model regarding the human annotation.”*
> >
> > **Response:**
> >
> > In our implementation, we use 3,600 human-annotated image pairs to train an SVM-based preference predictor, which then automatically labels additional pairs for training the reward model.
> >
> > To analyze the sensitivity of both the reward model and the restoration model to the amount of human annotation, we have added a new ablation study in Appendix D.5. Concretely, we vary the fraction of manual annotations used to train the SVM predictor:
> >
> > - **0%**: no manual data (we rely solely on the off-the-shelf HPSv2 without domain-specific fine-tuning);
> > - **50%**: half of the 3,600 annotated pairs;
> > - **100%**: all 3,600 annotated pairs (our default setting).
> >
> > For each setting, we report both reward quality (Human Consistency) and restoration quality. The quantitative results are summarized below (also provided as **Table 12 in Appendix D.5**):
> >
> > | Annotation Ratio | Human Consistency (↑) | MANIQA (↑) | MUSIQ (↑) | FID (↓) |
> > | ---------------- | --------------------- | ---------- | --------- | ------- |
> > | 0%               | 69.78%                | 0.6630     | 69.78     | 48.94   |
> > | 50%              | 83.21%                | 0.6689     | 73.32     | 45.90   |
> > | 100% (ours)      | 87.78%                | 0.6535     | 74.82     | 42.59   |
> >
> > We observe a clear positive correlation between the scale of human annotation and the quality of the reward model. Human Consistency improves from 69.78% → 83.21% → 87.78% as we increase the annotation ratio from 0% to 50% to 100%. This better alignment of the reward model with human judgments translates into improved restoration quality: MUSIQ increases from 69.78 to 74.82, and FID decreases from 48.94 to 42.59 (MANIQA remains at a comparable level).
> >
> > These results indicate that while the base HPSv2 already provides a reasonable perceptual prior, face-specific human annotations significantly enhance both the reward model and the final restored faces, and our framework continues to benefit from additional human supervision.
> >
> > ------
> >
> > ### W4. Significance of the text prompt in the reward model
> >
> > > *“The reward model introduces the additional text prompt. I wonder about the significance of this textual information.”*
> >
> > **Response:**
> >
> > These details were elaborated upon in **Appendix D.4** of our original submission. We provide a summary below:
> >
> > In our Face Reward Model, the text description is introduced to provide a **semantic anchor** for evaluating the restored face. Concretely, the image input allows the FRM to assess holistic visual quality (realism, details, aesthetics), while the text encodes desired semantic attributes (e.g., presence of glasses, age, hairstyle, accessories). The reward thus measures not only how realistic a face looks, but also whether its attributes are **consistent with the intended semantics**.
> >
> > We conducted an **ablation study** in **Appendix D.4**. In this ablation, we train two versions of the FRM:
> >
> > - **Image + Text:** the full face model used in our method.
> > - **Image + Null Text:** a control variant where the text input is replaced by a null string, i.e., the reward is image-only.
> >
> > We then evaluate how well each reward model agrees with human preferences on a manually annotated test set of 360 image pairs. As reported in Table 12 (Appendix D.4), using text improves Human Consistency from **85.01% → 87.78%**. This confirms that textual information helps the FRM align more closely with human judgments.
> >
> > Qualitatively, text is especially important when different plausible restorations differ in **semantic attributes** rather than overall realism. For example, in Fig. 11, the prompt explicitly mentions “glasses”. Both candidates (a) and (b) are visually plausible, but only (a) contains glasses. A text-conditioned reward correctly assigns a higher score to (a) and penalizes (b) as inconsistent with the description, whereas an image-only reward—which only sees two realistic faces—has difficulty capturing this mismatch.
> >
> > In summary, the text prompt is not strictly required for the reward model to function, but it provides a more precise and semantically grounded signal. This leads to better reward–human agreement and, in turn, more semantically faithful restoration results.

---

> > > ### Author Response · Authors · 2025-11-27
> > > **Paper 12356: Looking forward to your reply**
> > >
> > > Dear Reviewer,
> > >
> > > I hope this message finds you well. We have carefully clarified all your concerns and updated the manuscript accordingly.
> > >
> > > We are truly looking forward to discussing our work with you at a top-tier venue like ICLR. If there are any additional points or feedback you would like us to consider, please do not hesitate to let us know.
> > >
> > > We would greatly appreciate your reply soon, so we still have time to resolve any lingering concerns.

---

> > > > ### Comment · Reviewer_4Q9v · 2025-11-28
> > > > **Response to authors.**
> > > >
> > > > The response has addressed my concerns of W2-W4.
> > > >
> > > > However, for W1, I still have doubts about the necessity and effectiveness of the proposed reward loss. As shown in the response for Q1 of reviewer HVjR, the reward loss is not general for other architectures (only with slight improvement or obvious performance degradation regarding MUSIQ), e.g., CodeFormer.  I thus remain my original rating.

---

> ### Author Response · Authors · 2025-11-28
> **Clarification: ReFL Implementation in CodeFormer is Out of Scope**
>
> Thank you for your response. We have **never claimed** that our method can improve the restoration performance of all architectures. There are numerous types of restoration models, such as convolution-based networks, discrete codebook-based models (e.g., CodeFormer), Transformer-based models, Mamba-based models, diffusion-based models, etc., and no single approach can achieve performance gains across all of them. Regarding the experiment of transplanting DiffusionReward into the discrete-codebook-based CodeFormer, it was conducted merely to satisfy the curiosity of reviewer **HVjR**. In fact, this is entirely outside the scope of our research. Therefore, using it as a basis to reject our method is unprofessional.
> We have also clearly explained that CodeFormer is a discrete codebook approach, which is fundamentally different from probabilistic generative models such as diffusion.
>
>
> The claim in our paper is specifically that it improves the restoration performance of diffusion-based models. The evidence is as follows:
>
> Firstly, our method is called **DiffusionReward**, which explicitly points to the Diffusion Model.
>
> Second, we explicitly stated in the **Intro.** that our work is fundamentally designed to address the specific limitations of diffusion-based restoration methods, rather than claiming to be a universal solution for all architectures. As described in the **Intro:**
>
> *"However, these pre-trained diffusion models typically undergo training using images from general domains, which lack an adequate amount of face-specific prior knowledge. This deficiency frequently gives rise to restored facial images that are short of detailed features."*
>
> In the **methods Sec.**, Our method is theoretically modeled on Diffusion models, leveraging the stochasticity in the denoising process to implement ReFL.
>
> Finally, in the text-to-image domain, ReFL is inherently designed for Diffusion methods, as seen in works like ImageReward [1], DRaFT [2], and R0 [3]. Traditional architectures like CodeFormer lack the stochastic nature required for these methods to be effective, which was precisely the purpose of conducting the CodeFormer experiment—to verify this distinction.
>
> [1] Jiazheng Xu, Xiao Liu, Yuchen Wu, Yuxuan Tong, Qinkai Li, Ming Ding, Jie Tang, and Yuxiao Dong. Imagereward: learning and evaluating human preferences for text-to-image generation. In Advances in Neural Information Processing Systems (NeurIPS), 2023
>
> [2] Kevin Clark, Paul Vicol, Kevin Swersky, and David J Fleet. Directly fine-tuning diffusion models on differentiable rewards. arXiv preprint arXiv:2309.17400, 2023.
>
> [3] Yihong Luo, Tianyang Hu, Weijian Luo, Kenji Kawaguchi, and Jing Tang. Rewards are enough for fast photo-realistic text-to-image generation. arXiv preprint arXiv:2503.13070, 2025

---

### Official Review · Reviewer_jFAM · 2025-10-31

**Soundness:** 3
**Presentation:** 3
**Contribution:** 4
**Rating:** 6
**Confidence:** 3

**Summary:**

This paper introduces DiffusionReward, a novel Reward Feedback Learning (ReFL) framework for blind face restoration (BFR). The work addresses critical limitations of diffusion-based BFR methods, such as insufficient facial details and poor identity consistency, by leveraging a carefully designed Face Reward Model (FRM) and dynamic optimization constraints. The idea of integrating ReFL into BFR is innovative, and the experimental results demonstrate state-of-the-art performance on both synthetic and real-world datasets. The paper is well-structured, and the methodology is technically sound.

**Strengths:**

Novelty and Significance:

1) This is the first work to adapt ReFL for BFR, bridging a gap between generative alignment and restoration tasks. The dynamic FRM update strategy effectively mitigates reward hacking, a common pitfall in reward-based optimization.

2) The hybrid annotation pipeline (combining human labels with SVM-based automation) for FRM training is resource-efficient and scalable.

Technical Rigor:

1) The framework incorporates multiple loss terms (reward loss, structural consistency, weight regularization) to balance perceptual quality, identity preservation, and generative diversity. Ablation studies thoroughly validate each component’s contribution.

2) Experiments cover diverse settings: two base models (DiffBIR and OSEDiff), synthetic (CelebA-Test) and wild datasets (LFW/WebPhoto-Test), and 11 metrics. The consistent improvements highlight generalizability.

**Weaknesses:**

Clarity of FRM Training Details: The description of the SVM-based automated annotation (Section 3.1) is concise but lacks critical specifics (e.g., kernel choice, feature normalization). Please add a brief summary in the main text or refer to Appendix A.1 for clarity.

Limitations of Generalizability: The framework is only validated on diffusion-based models (DiffBIR/OSEDiff). The ablation in Appendix G.1 shows limited gains when applied to GFPGAN (GAN-based). This should be explicitly discussed as a limitation.

Computational Cost: While truncated backpropagation (N=1) reduces memory usage, the dynamic FRM updates and multi-constraint optimization still incur overhead. A brief discussion of training efficiency would strengthen the practicality claims.

**Questions:**

Please refer to the paper weakness

---

> ### Author Response · Authors · 2025-11-23
> **Response to Reviewer jFAM**
>
> Thank you to the reviewer for acknowledging our work and for the careful review. Regarding your concerns, we have revised and uploaded the manuscript, with all changes highlighted in blue for easy reference. Next, we address the reviewer's concerns point-by-point.
>
> ### W1. Clarity of FRM training details
>
> > *“Clarity of FRM Training Details: The description of the SVM-based automated annotation (Section 3.1) is concise but lacks critical specifics (e.g., kernel choice, feature normalization). Please add a brief summary in the main text or refer to Appendix A.1 for clarity.”*
>
> **Response:**
>
> We have added a relevant sentence in **Section 3.1** of the main paper to remind readers that the key details of the SVM can be found in **Appendix A.1**.
>
> ------
>
> ### W2. Limitations of generalizability beyond diffusion-based models
>
> > *“Limitations of Generalizability: The framework is only validated on diffusion-based models (DiffBIR/OSEDiff). The ablation in Appendix G.1 shows limited gains when applied to GFPGAN (GAN-based). This should be explicitly discussed as a limitation.”*
>
> **Response:**
>
>  Thank you for your reminder. The limitations of our proposed method on non-diffusion architectures (e.g., GFPGAN) have been incorporated into the Limitations section. Furthermore, the Limitations section has been moved into the main body.
>
> ------
>
> ### W3. Computational cost and training efficiency
>
> > *“Computational Cost: While truncated backpropagation (N=1) reduces memory usage, the dynamic FRM updates and multi-constraint optimization still incur overhead. A brief discussion of training efficiency would strengthen the practicality claims.”*
>
> **Response:**
>
> We added a detailed discussion on the training efficiency of DiffusionReward in **Appendix E** . To briefly summarize our key insights: our approach is a form of post-training on a base model that achieves permanent performance gains with only a small number of iterations. Consequently, the ReFL training time required is far less than the original pre-training time of the base model, making it a highly training-efficient method.
>
> For more detailed quantitative data and discussion regarding training efficiency, please refer to **Appendix E** of our revised manuscript.

---

> > ### Author Response · Authors · 2025-11-27
> > **Paper 12356: Looking forward to your reply**
> >
> > Dear Reviewer,
> >
> > Thank you again for your careful and constructive review.
> >
> > We have carefully incorporated all of your helpful minor suggestions into the revised manuscript. Regarding your very valid concern about training overhead, we have now added a dedicated discussion in the new version (Appendix E).
> >
> > We sincerely hope these changes and clarifications have fully addressed your concerns. If anything still remains unclear or could be further improved, we would be truly grateful for even a brief final note in the remaining time.
> >
> > Looking forward to hearing your thoughts.

---

### Author Response · Authors · 2025-11-23
**Summary of Revisions to the DiffusionReward Paper**

We sincerely thank all reviewers for their constructive feedback. Following the suggestions, we have made substantial revisions to improve the clarity, completeness, and contributions of the paper. **All modifications are highlighted in blue in the revised version.** The main updates are as follows:

### Related Work

- We have added two references “Face Super-Resolution Guided by 3D Facial Priors” and “Rethinking Deep Face Restoration.”
- We clarified the distinct challenges of applying ReFL to restoration.


------

### Method (Sec. 3.2 & Sec. 3.3)

- We clarified that Eq. (1) describes the *standard* latent diffusion noise-prediction objective used only to pre-train the base BFR models, and is **not** the loss used in our ReFL stage.
- In the ReFL section, we now explicitly state that our fine-tuning runs the full reverse denoising chain in the forward pass, but applies truncated backpropagation with $N=1$ on a reward-driven loss (plus structural and regularization terms) defined on the final restored image.

------

### Experiments

- We moved the gradient truncation ablation (originally Appendix D.2) into the main experimental section (Sec. 4.3 and Table 5).
- We added **training dynamics plots** (Fig. 7 in the revised version), showing the evolution of structural losses and reward values during ReFL fine-tuning, confirming stable convergence behavior.
- We moved the user study in the appendix to the main paper.

------

### Limitations

- We moved and strengthened the Limitations section in the main body, explicitly discussing the more limited gains on non-diffusion architectures.

------

### Appendix

- We added a table of contents for the appendix to make the overall structure clear.

- We added a new ablation on the amount of human annotation (App. D.5), where we vary the ratio of manual pairs (0%, 50%, 100%) used to train the SVM annotator and report both Human Consistency and restoration metrics, showing steady gains as annotation increases.
- We added a new experiment on FRM stability under dynamic updates (App. D.3), reporting Human Consistency at multiple update iterations and showing that the FRM consistently maintains $>83$% agreement with human judgments.
- We added unbiased qualitative comparisons (CelebA-Test images 00000000–00000019) in Appendix H.2 .
- Following the reviewer’s suggestion, we added a sensitivity analysis of the KL weight regularization coefficient $\lambda_{\text{reg}}$ (Variant 1 without Reward Model).
- We added a quantitative analysis of training cost and inference latency (Table 14), reporting training hours, per-image inference time, and key metrics for both base models and our fine-tuned versions.

---

### Meta-Review · Area_Chair_gevT · 2026-01-06

**Summary:**

This paper studies blind face restoration (BFR) and proposes a reward feedback learning framework, DiffusionReward, for diffusion-based BFR models. The method collects a dataset of degraded face images and their restoration results, obtains preference annotations through a hybrid pipeline that includes human labeling and an SVM-assisted labeling stage, and trains a Face Reward Model (FRM). The FRM is then used to provide differentiable reward signals to guide the fine-tuning of diffusion-based restoration backbones (e.g., DiffBIR and OSEDiff). The framework also introduces multiple constraints, including structural consistency and weight regularization, and proposes a dynamic reward model update strategy aimed at mitigating reward hacking. The paper reports improvements on both synthetic and real-world benchmarks with extensive quantitative and qualitative evaluations.

**Reviewer Concerns:**

Reviewer jFAM (initial score 6, positive): This reviewer views the work as an innovative application of reward feedback learning to BFR and appreciates the dynamic FRM update to mitigate reward hacking, as well as the hybrid annotation strategy. The main concerns are relatively minor: more details on SVM labeling, clearer discussion of limitations outside diffusion models, and a brief discussion of computational overhead. The authors responded by adding pointers to appendix details, moving limitations into the main paper, and adding training cost discussion. The reviewer did not follow up in the discussion. From the AC perspective, these points appear addressable and are not decisive.


Reviewer 4Q9v (initial score 4, negative): This reviewer raises the most central concern: the paper’s core contribution is introducing reward feedback into BFR, yet the ablations do not convincingly establish the necessity and unique benefit of the reward loss, since the gains can be largely attributed to structural consistency and weight regularization in their reading. The reviewer also questions the significance of the truncation setting (N=1) and asks for sensitivity analysis regarding human annotation scale and the role of text prompts. The authors provided additional sensitivity analysis and clarifications, and the reviewer acknowledged that W2–W4 were addressed, but maintained the original negative rating due to unresolved doubts about W1, i.e., whether the reward loss is truly necessary and effective, and whether its benefit generalizes beyond diffusion models.


Reviewer nDxx (initial score 6, positive): This reviewer is overall positive and emphasizes the potential impact of bringing ReFL to BFR, as well as the careful construction of the FRM and data curation. The concerns include potential cherry-picking in qualitative results, clarity of novelty relative to existing ReFL work in generation, and minor presentation issues. The authors added non-cherry-picked qualitative results and clarified novelty in the revision. From the AC perspective, these concerns are largely presentation or completeness issues and appear adequately addressed.

Reviewer HVjR (initial score 4, negative): This reviewer’s key concerns are the lack of deeper analysis on reward accuracy and reward sparsity or noise sensitivity, lack of convergence and training dynamics analysis, and missing cost analysis. They also ask whether the reward learning paradigm is architecture-agnostic or diffusion-specific, noting strong non-diffusion BFR methods (e.g., CodeFormer) and asking for clarity on scope. The authors responded with additional human-consistency measurements, training dynamics plots, and training cost tables, and discussed scope and applicability beyond diffusion. However, the reviewer’s broader concern about what exactly the reward is capturing and how robustly it captures it remains important.

**Reviewer Scores:**

After reading the manuscript, the reviews, and the rebuttal and discussion, my assessment is that the paper contains ideas and strong experimental effort, but still has unresolved issues around positioning, conceptual clarity, and the necessity of the proposed reward framework, which collectively prevent acceptance in the current version.

The paper’s framing emphasizes bringing ReFL into diffusion-based BFR. However, at a conceptual level, the approach largely reduces to training a new reward model and optimizing against it, together with several auxiliary losses. The paper does not provide a sufficiently clear discussion of how this reward formulation relates to, or differs from, existing objectives commonly used in BFR and restoration pipelines, such as perceptual losses, feature-space losses, ranking losses, and other preference-based objectives. Even if the empirical gains are real, the paper does not articulate what new principle this introduces beyond a learned objective, nor what practitioners should take away when choosing between a learned reward and established loss designs. The paper argues that dynamic reward model updates help prevent reward hacking. However, the paper does not clearly explain how this differs from mechanisms in GAN training.

If the intended scope is diffusion-only, the paper needs to more rigorously justify this scope limitation and explain what properties of diffusion make the approach appropriate.

While the paper reports human-consistency numbers, the labeling pipeline and the role of SVM-assisted annotation raise questions about what signal is ultimately captured. In particular, the use of multiple low-level metrics in the labeling pipeline, without a clearer semantic or identity-grounded rationale, creates ambiguity: is the reward primarily measuring low-level appearance quality, identity preservation, semantic faithfulness, or a mixture? The paper does not sufficiently clarify the target concept being optimized, and the SVM-assisted pipeline risks biasing the reward toward the chosen proxy metrics. This aligns with concerns about reward reliability and reward signal quality raised by negative reviewers.

Overall, I agree with the broad theme from the two negative reviewers: the paper presents an interesting framework and empirical improvements, but it does not yet clearly demonstrate the unique necessity and advantage of the reward model relative to established alternatives, nor does it provide sufficiently rigorous conceptual grounding and positioning.

---

### Decision · Program_Chairs · 2026-01-26

Reject